

# Isotopic exchange on the diurnal scale between near-surface snow and lower atmospheric water vapor at Kohnen station, East Antarctica

François Ritter[1,2], Hans Christian Steen-Larsen[2,3], Martin Werner[1], Valérie Masson-Delmotte[2], Anais Orsi[2], Melanie Behrens[1], Gerit Birnbaum[1], Johannes Freitag[1], Camille Risi[4], and Sepp Kipfstuhl[1]

[1]Alfred-Wegener-Institut (AWI) Helmholtz-Zentrum für Polar und Meeresforschung, Bremerhaven, Germany.
[2]Laboratoire des Sciences du Climat et de l'Environnement (LSCE), IPSL/CEA-CNRS-UVSQ, Saclay, Gif-sur-Yvette, France.
[3]Center for Ice and Climate, Niels Bohr Institute, University of Copenhagen, Copenhagen, Denmark.
[4]Laboratoire de Météorologie Dynamique (LMD), IPSL/CNRS-UPMC, Paris, France.

*Correspondence to:* François Ritter (fritte2@uic.edu)

**Abstract.** Quantifying the magnitude of post-depositional processes affecting the isotopic composition of surface snow is essential for a more accurate interpretation of ice core data. To achieve this, high temporal resolution measurements of both lower atmospheric water vapor and surface snow isotopic composition are required. This study presents the first continuous measurements of water

5 vapor isotopes performed in East Antarctica (Kohnen station) from December 2013 to January 2014 using a laser spectrometer. During our monitoring period, the synoptic variability of the water vapor isotopic composition is found to be low compared to the diurnal cycle and we therefore concentrate our study on interaction between the isotopic composition of the vapor and the snow surface on a diurnal timescale. The peak-to-peak amplitude of the snow surface isotopic composition over 24 h

10 reaches 3 ‰ for $\delta D$, in phase with the diurnal variations of $\delta D$ in surface vapor, which itself has an amplitude of 36 ‰. A simple box model treated as a closed system has been developed to study the exchange of water molecules between an air and a snow reservoir. In the vapor, the simulations show too much isotopic depletion compared to the observations. Mixing with other sources (wind advection, free troposphere) has to be included in order to fit the observations. At the snow surface,

15 the simulated isotopic values are close to the observations with a snow reservoir of ∼5 mm depth (range of the snow sample depth). Our analysis suggests that vapor-snow exchanges can no longer be considered insignificant for the isotopic composition of near surface snow in central Antarctica.



## 1 Introduction

Thanks to the design of mass spectrometers and their application to water stable isotopes since the
1950s, precipitation has long been sampled for laboratory stable isotope analyses in order to trace
atmospheric processes related to the hydrological cycle (e.g., Dansgaard, 1964). Past changes in
precipitation isotopic composition have also been investigated using a variety of natural archives.
Among these, ice cores form one of the most direct records of the isotopic composition of past
precipitation. In Antarctica, water stable isotope measurements are central for past climate recon-
structions from these ice cores, through atmospheric distillation processes connecting temperature,
condensation and isotopic composition (Masson-Delmotte et al., 2008, e.g.,). However, the rela-
tionship between precipitation isotopic composition and climate is complex, as it is affected by
fractionation taking place at each phase transition, at evaporation, during atmospheric transport, and
condensation.

While spatial relationships have been documented using surface snow data (Masson-Delmotte
et al., 2008), only few studies have examined the drivers of Antarctic precipitation isotopic compo-
sition variability at the meteorological or shorter time scales (Fujita and Abe, 2006; Schlosser et al.,
2004). In this study we focus on these timescales. The climatic controls on precipitation isotopic
composition have been investigated under different climatic contexts thanks to the implementation
of stable water isotopes into Atmospheric General Circulation Models (AGCM) (Joussaume et al.,
1984, e.g.,). Classically, the mean precipitation isotopic composition simulated by atmospheric mod-
els is directly compared to ice core data, thereby ignoring potential post-deposition processes which
may transform the initial precipitation signal in the upper part of the firn. In this study we aim at
investigating one such post-deposition process: the exchange between atmospheric water vapor and
snow.

The possibility to monitor isotopic exchanges between surface snow and low level atmospheric
water vapor has emerged thanks to recent technological development. New laser spectrometers have
been released by two companies (Los Gatos Research Inc. and Picarro Inc.) in the 2000s, creating a
substantial advance within the field of water isotope research. These analyzers are able to perform
continuous and in-situ measurements of the humidity mass ratio (hereafter humidity mixing ratio)
and the stable water vapor isotope concentrations. Prior to this advance, water vapor could only be
collected using tedious cold trapping methods (e.g., Jacob and Sonntag, 1991; Steen-Larsen et al.,
2011; Angert et al., 2008), deployed until sufficient amounts were collected to allow subsequent
transfer to vials and later mass spectrometer analyses. Such a task was almost impossible to perform
routinely (e.g., Schwarz et al., 1998).

The first implementation of continuous in-situ isotopic monitoring of surface water vapor isotopic
composition above an ice sheet was achieved at NEEM, NW Greenland, during summer field sea-
sons 2010-2012 (Steen-Larsen et al., 2013, 2014). In parallel, repeated sampling of surface snow
was also implemented for laboratory isotopic analyses. The combined snow and vapor datasets re-



vealed changes in the isotopic composition of surface snow in response to synoptic changes in the atmospheric water vapor isotope values, and exhibited a strong diurnal variability in near-surface water vapor isotopic composition. Over several days and weeks, parallel variations between the surface snow isotopic composition and the near-surface vapor isotopic composition were identified in between snowfall events. This was surprising, as the snow isotopic composition was expected to be

controlled by the isotopic composition of precipitation. The variations were interpreted to reflect interactions between the snow surface and the lower atmosphere occurring during surface snow metamorphism. This new finding has potential implications for the interpretation of ice core records, and for the comparison of ice core data with atmospheric model results. It is unclear however, whether surface air-snow exchanges could lead to significant changes in the snow isotopic composition at

colder sites, such as those of the Antarctic Plateau.

Here, we report the first surface vapor isotope measurements performed above the Antarctic ice sheet. These measurements were performed at the German Kohnen station, a deep ice coring site with intermediate temperature and a moisture level high enough in the summer for making accurate measurements of the water vapor isotopic composition. After a brief overview of the Kohnen station

environment (Sect. 2), this article details the successful implementation of continuous measurements of water vapor isotopic composition during the months of December 2013 and January 2014. Section 3 (material and methods) describes our protocol for water vapor data processing, and reports the accuracy of the data. We also report the parallel surface snow sampling over 35 hours (hereafter SSDC experiment for Snow Surface Diurnal Cycles) and the subsequent laboratory measurements.

We introduce simulations performed with two atmospheric general circulation models (LMDZiso and ECHAM5-wiso) equipped with stable water isotopes and used for comparison with our observational data. Atmospheric general circulation models equipped with water stable isotopes are commonly used to quantify the relationships between simulated climate parameters and Antarctic precipitation water stable isotopes, at different time scales. Here, we therefore explore the ability of

the two models to capture the diurnal and synoptic variability observed during our field campaign. In Sect. 4, we present observed and simulated values, first for the day to day variability and then we focus on the diurnal cycles. We compare the diurnal variability of the isotopic composition of the water vapor and of the very first layer of surface snow through a box model. The last section summarizes our key conclusions and perspectives.

## 85  2   Kohnen station environment

The German Kohnen station [75°00'S, 0°04'E] is located on the Antarctic Plateau in Dronning Maud Land, 550 km from the South Atlantic coast line and 2892 m above sea level. Near the station, the surface elevation has a gentle slope of $\sim$1.3$\pm$0.3 m.km$^{-1}$ with a direction of $\sim$61 °. This place is characterized by katabatic winds ($\sim$8 m.s$^{-1}$) on diurnal time scale, which form around 3 h and





vanish around 15 h (UTC time) (Van As et al., 2005). The katabatic regime can be interrupted by the influence of synoptic systems responsible for the majority of the rare precipitation events at Kohnen, bringing small precipitation amounts during the summer season. From December 2013 to January 2014, five snowfall events were observed and no significant snow accumulation was detected from daily measurements of snow surface height within a precision of ∼1 mm.

The mean temperature for the month of January 2014 was -25 °C, similar to the climatological average (-25±2 °C, 1998 to 2013). Both temperature values are based on hourly data from the permanent Automatic Weather Station at Kohnen (hereafter AWS9, described in Sect. 3.1). The diurnal cycle in the air temperature has an amplitude of 10 °C (Fig. 3), which is consistent with the earlier observations of Van As et al. (2005) at Kohnen. During clear sky conditions, the net radiation

at the surface is predominantly positive during the day (i.e., a dominant energy gain by shortwave emissions from the sun despite a high albedo of the snow surface) and negative during the night (i.e., a net loss by longwave emissions from the surface), imprinting a strong diurnal cycle to the surface temperature.

The surface of the ice sheet around Kohnen is characterized by the presence of large sastrugi,

created by wind redistribution and sublimation of snow, hence producing considerable variability in the snow surface age, origin, density and isotopic composition. In particular, very hard dunes sticking up above the mean surface level may be half a year old and are expected to have a different isotopic composition from the freshly deposited snow.

## 3  Material and methods

### 3.1  Weather observations


Weather observations are reported every hour and precipitation events are labeled 'snowfall' or 'light snowfall'. A snowfall event leaves a visible accumulation on flat surfaces (for example transport boxes) whereas during a light snowfall event no visible accumulation is observed.

Two Automatic Weather Stations (AWS) were also installed at Kohnen. AWS9 is permanent and

performs hourly measurements since ∼1998, whereas AWS 13/14 was only temporarily installed at Kohnen for the summer season and performed measurements every minute. Both were measuring the air pressure, temperature, relative humidity, wind speed and wind direction at 2 m above the snow surface. Their relative locations are indicated in Fig. 1.

### 3.2  Water vapor sampling system

Measurements of water vapor isotopes in the near surface atmosphere were performed from 2013/12/17 to 2014/01/21 using a Los Gatos Research Inc. analyzer (hereafter simply analyzer), type DTL-100. It continuously measured the humidity mixing ratio and the relative composition of two stable water isotopes: $R_{^{18}O} = [^1H_2^{18}O]/[^1H_2^{16}O]$ and $R_D = [^1H^2H^{16}O]/[^1H_2^{16}O]$ (Baer et al., 2002). We will use





throughout this paper the standard $\delta$ notation in ‰:

$$\delta^* = \left( \frac{R_*}{R_{\mathrm{VSMOW}}} - 1 \right) \times 1000 \tag{1}$$


where $\delta^*$ stands for $\delta D$ or $\delta^{18}O$, and $R_{\mathrm{VSMOW}}$ is the relative composition of the Vienna Standard Mean Ocean Water. The instrumental temporal resolution is 2 Hz, but we report measurements averaged over 11 minutes to increase the signal to noise ratio. The precision decreases with humidity, leading us to exclude all measurements performed below 500 ppmv.

The analyzer was calibrated using a stream of water vapor with known constant isotopic composition generated by the Water Vapor Isotope Standard Source (WVISS, Los Gatos Research Inc.). The WVISS allowed control over the amount of dilution of the vapor stream resulting in vapor stream of adjustable humidity level. A working standard was created at the beginning of the campaign by melting surface snow and subsequently stored in a sealed glass container. Samples were taken from the working standard every two weeks for later laboratory isotopic analysis to check for stability. No significant drift was observed. The working standard was calibrated against VSMOW, SLAP and GISP at the Alfred Wegener Institute, Bremerhaven (hereafter AWI). Its isotopic composition was: $\delta^{18}O = -44.44 \pm 0.03$ ‰, and $\delta D = -345.5 \pm 0.1$ ‰.

Figure 1 shows the location of the instrument compared to the main wind direction ($\sim 61$ ° true north) and its distance to the base. The large clean area prevents any perturbations from local human activity. Measurements were performed at three different heights above the snow surface: 0.2, 0.9 and 3 m. Insulated and heated copper tubes were used to suck in air to the analyzer following the setup of Steen-Larsen et al. (2013). An air-filter and a snowfall protection were placed on each inlet to prevent sucking in snow crystals. The three copper tubes and the WVISS were connected to a device called the Multiport Input Unit (Los Gatos Research Inc.). This device was controlled by the analyzer to switch the valves and alternate between the inlet measurements and WVISS calibrations. Dry air was flushed through the system to check for leaks. The order of measurements during the campaign was the following: 12 minutes calibration followed by three cycles of 33 minutes, measuring 11 minutes at each inlet.

## 3.3   Calibrations

The calibration protocol follows Steen-Larsen et al. (2013). In short, one calibration is applied to the humidity mixing ratio and then three types of calibrations of the measured water vapor isotope signal are performed: instrumental humidity-isotope response calibration, VSMOW-SLAP calibration and drift correction. More details about the calibration procedures are given in the supplementary material.

The humidity mixing ratio is calibrated against the relative humidity measured by the AWS9. We first apply the Anderson correction to the AWS9 relative humidity (Anderson, 1994) and then we convert the relative humidity into the humidity mixing ratio $q$ using the surface pressure (Goff-




Gratch equation with respect to ice) and air temperature at 2 m height. We finally calculate a fit from
AWS9 to analyzer $q$, with a 2nd order polynomial to get the function correcting the humidity mixing
ratio measured during the campaign. This function is $f(q) = a + b \times q + c \times q^2$ with a = 80±80, b
= 0.59±0.14 and c = (0.23±0.06)×10$^{-3}$. We have checked the linear relationship between $\delta D$ vs $q$
(shown in Table 5) with/without this calibration and conclude that the slope is not sensitive.

The instrumental humidity-isotope response calibration was obtained by varying the humidity
level of the vapor stream produced by the WVISS while measuring the same standard. As the water
vapor isotopic composition generated by the WVISS is assumed to be constant, the observed vari-
ation of the water vapor isotopes can be attributed to the instrumental humidity response. Five hu-
midity calibrations were performed during the campaign (2013/12/06, 12/14, 12/28 and 2014/01/13,
01/28). For each of them, we continuously generated a stream of water vapor from the "Work-
ing Standard water", using the WVISS during ∼20 hours. We forced the humidity to vary from
∼4000 ppmv to 300 ppmv by changing the amount of dilution. The humidity isotope response in-
between calibration periods is assumed to vary linearly.

The VSMOW-SLAP calibration is carried out by measuring vapor from the WVISS generated
when evaporating standards of known isotopic composition referenced against the VSMOW-SLAP
scale. Four different water standards (NZE, NEEM, TALOS, OC3) with a known isotopic composi-
tion were brought to Kohnen station in 40 glass bottles of 10 cl. During a VSMOW-SLAP calibration,
each standard was vaporized and measured during 15+15 minutes at two different humidity levels,
which were used to check the accuracy of the humidity correction. This calibration lasted ∼ 6 hours
and has been reproduced four times on 2013/12/06, 2013/12/15 and 2014/01/08, 2014/01/22. Very
small variations in the VSMOW-SLAP calibration slope were observed (see Table 1) and we have
therefore simply calculated the mean value to obtain our conversion slopes : $\alpha_{\delta D} = 1.43 \pm 0.02$ and
$\alpha_{\delta^{18}O} = 1.00 \pm 0.03$ (uncertainties are the standard error of the mean value).

Finally the measured isotopic value is corrected for the drift by measuring vapor generated by
the WVISS when evaporating the prepared working standard. This measurement is performed every
111 minutes during 12 minutes and linear interpolation is assumed between each 'drift'-correction
measurement.

### 3.4 Precision and accuracy of measurements

We use the following notation to describe the error propagation, with $\delta*$ standing either for $\delta D$ or
$\delta^{18}O$.

1. The raw isotopic composition (direct output from the analyzer without any corrections) aver-
   aged over 11 minutes is $\delta^*_{\text{raw}} \pm d\delta^*_{\text{raw}}$ with $d\delta^*_{\text{raw}}$ the standard error associated with the mean
   value.





2. The humidity isotope response associated to $\delta^*_{\text{raw}}$ is $\Gamma_* \pm d\Gamma_*$ with $d\Gamma_*$ the uncertainty associated with the humidity correction. $d\Gamma_*$ is for a given $q$ and a given time the absolute difference between two humidity isotope responses from two consecutive humidity calibrations. $d\Gamma_*$ is therefore taken as the maximum possible error on the humidity correction.

3. The slope conversion to the VSMOW-SLAP international scale is $\alpha_* \pm d\alpha_*$ with $d\alpha_*$ the uncertainty associated with the slope. $d\alpha_*$ is the standard error of the mean value of the slopes from the 4 different VSMOW calibrations.

4. The drift correction is $\mu_* \pm d\mu_*$ with $d\mu_*$ the uncertainty associated with the drift correction. $d\mu_*$ has been estimated at the end of the campaign by performing an extra 24 hours calibration with a stable humidity.

5. The corrected isotopic composition is $\delta^*_{\text{corr}} \pm d\delta^*_{\text{corr}}$ with $d\delta^*_{\text{corr}}$ the final uncertainty containing both the precision and the accuracy on the corrected measurements.

We apply the three corrections (humidity isotope response, conversion to the VSMOW-SLAP scale, drift removal) to calculate the corrected isotopic composition at a given time:

$$\delta^*_{\text{corr}} = \alpha_* \times (\delta^*_{\text{raw}} - \Gamma_*) - \mu_*$$

We obtain a final uncertainty on the corrected isotopic composition by applying an error propagation calculation, assuming no correlation between the three corrections:

$$
(d\delta^*_{\text{corr}})^2 = \overbrace{\alpha_*^2 \times \left((d\delta^*_{\text{raw}})^2 + (d\Gamma_*)^2\right) + (d\mu_*)^2}^{P_*^2} \\
+ \underbrace{(d\alpha_*)^2 \times (\delta^*_{\text{raw}} - \Gamma_*)^2}_{A_*^2}
$$

With $A^*$ standing for the accuracy of the measurements and $P^*$ standing for the precision of the measurements. We have attributed the part $A^*$ to the accuracy because the uncertainty on the VSMOW correction will affect the mean value of the data over the campaign. Each correction depends on the time of the measurement (the drift varies through time, the humidity isotope response as well) and Table 2 summarizes the different orders of magnitude of the parameters with the estimated precision/accuracy.

We estimate a precision of the measurements of 3.0 ‰ for $\delta D$ and of 0.9 ‰ for $\delta^{18}O$. We estimate an accuracy of the measurements of 11 ‰ for $\delta D$ and of 2.5 ‰ for $\delta^{18}O$. When focusing on the mean diurnal cycle, we will get a higher precision by calculating hourly averages over 18 days (Sect. 4.2).



### 3.5 Surface snow sampling and analysis

In order to detect any snow accumulation or depletion (due to snowfall events or wind drift), 100 thin wood sticks were distributed every meter along a 100 m transect in a clean area. Over a duration of 50 days - from 2013/12/07 to 2014/01/27 - no accumulation or depletion was detected through daily measurements with a precision of one millimeter.

The Snow Surface Diurnal Cycle experiment (SSDC) was devoted to the detection of a diurnal cycle in the isotopic composition of surface snow. To this aim, we sampled the surface snow hourly for a 35-hour period during clear sky weather, from 2014/01/08 to 2014/01/10 (as it is shown in Fig. 2 indicated by the SSDC label). Keeping in mind the large variability in surface snow isotopic composition, we selected three areas with consistent surface snow texture (patch 1, hard snow; patch 2, medium snow; patch 3, soft snow), and took 5 adjacent samples from each patch every hour. The surface of the snow (upper $\sim$ 2-5 mm) was scraped into a plastic bag, which was sealed and shipped for subsequent isotope measurements at the AWI. Snow samples were measured at AWI using two water isotope analyzers Picarro, type L1102-i and L2120-i. The protocol followed Geldern and Barth (2012).

### 3.6 Atmospheric simulations

ECHAM5-wiso (Werner et al., 2011) is the isotopic version of the atmospheric general circulation model ECHAM5 (Roeckner et al., 2003). Simulations from ECHAM5-wiso are nudged to the European Center for Medium-Range Weather Forecasts (ECMWF) ERA-interim reanalyses data (Berrisford et al., 2011) using 6-hourly pressure, temperature, divergence and vorticity fields. Sea surface temperatures and sea ice coverage are derived from the ERA-interim data set too. For our purpose this model has been run with a high vertical and horizontal T106L31 resolution (31 levels, 1.1 ° in longitude × 1.1 ° in latitude). The lowest model level has been selected followed by a bi-linear interpolation of nearby model grid points to the location of Kohnen base [75°00'S, 0°04'E].

LMDZ5Aiso (hereafter LMDZiso) is the isotopic version of LMDZ5A, part of the atmospheric general circulation model IPSL-CM5A used in the Coupled Model Intercomparison Project (CMIP5) (Risi et al., 2010). For our purpose this model has been run with a vertical resolution of 39 levels and a horizontal resolution of 2.5 ° x 3.75 °. Simulations are constrained by sea surface temperature data from the National Centers for Environmental Prediction (NCEP) and nudged to the 6-hourly ECMWF analyses using only the wind fields.

Both ECHAM5-wiso and LMDZiso simulations have been started in 1979 and are equilibrated for our study period.

Some selected outputs from both models are calculated at a precise height, for example at 10 m for the wind speed and wind direction, and at 2 m for temperature. However, the reader should





notice that here we use the simulated humidity for the first vertical model level whereas the in-situ observations are close to the surface.

## 4  Results and discussion

We present the observed and simulated hourly variations of temperature, humidity mixing ratio, deuterium and d-excess in Fig. 2, as a function of time. Section 4.1 will be devoted to the study

of the day-to-day variability over the period of the campaign in both air observations at 3 m and simulations from ECHAM5-wiso and LMDZiso. Section 4.2 will focus on the diurnal scale, showing subtle differences between the 0.2 and 3 m inlets in the mean value of 18 selected days (labeled as horizontal orange bars in Fig. 2) and comparing it with LMDZiso/ECHAM5-wiso outputs. Finally, Sect. 4.3 will study the diurnal cycle in the snow surface, by comparing the results from the snow

surface samples collected during the SSDC experiment (labeled as the horizontal purple bar in Fig. 2) with isotopic simulations from a snow surface-air model running as a closed system.

### 4.1  Observed and simulated day-to-day variability

In order to estimate the magnitude of the day-to-day variability, we have first removed days which contain more than 8 hours of data gap (on 12/16, 12/28, 12/29, 01/13, 01/14, 01/17, 01/18 and 01/21)

and then calculated daily mean values from the 29 remaining days. Table 3 presents the average and the standard deviation over the 29 daily mean values for the observations and the model outputs. The mean humidity mixing ratio measured from the top inlet is 1100 ppmv, with a maximum measured at 2200 ppmv, on 2013/12/22, at 15 h UTC. It coincides with the highest temperature, reaching - 15.5 °C at 17 h UTC on the same day. By contrast, the driest and coldest conditions are encountered

on 2014/01/19, at 2 h UTC, and is estimated at 150 ppmv for the humidity mixing ratio (therefore outside of the range were our calibration performance is satisfying) and measured at -35.8 °C for the air temperature. Deuterium mean value at 3 m is -410 ‰, with a range of variation from -360 ‰ (on 12/18 at 17 h UTC) to -470 ‰ (on 01/21 at 3 h UTC).

We first compare the model performances for these diurnal mean values. ECHAM5-wiso cor-

rectly simulates these mean conditions for the temperature, humidity mixing ratio and deuterium while LMDZiso produces too warm and wet near-surface values and presents an significant offset of ∼130 ‰ for deuterium (Fig. 2), as expected from the lack of distillation associated with a warm and wet bias.

Observations depict a day-to-day variability of 200 ppmv for the humidity mixing ratio, 9 ‰ for

deuterium and 4 °C for temperature (Fig. 2). Both models underestimate the variability of temperature and humidity mixing ratio. ECHAM5-iso also overestimates the variability of deuterium. One explanation concerning temperature could be that both models fail to capture the very cold events observed on 12/30, 01/19 and 01/20. They also underestimate the variations of temperature and hu-



midity mixing ratio observed over several days of relatively clear sky from 12/20 to 12/24, with

an amplitude in $T$ ($q$) of 0.9 °C (250 ppmv) for ECHAM5-wiso against 4.2 °C (490 ppmv) for the observations. The deuterium has a high day-to-day variability in ECHAM5-wiso because of the simulation of very depletion events which strongly deviate from the observations. For example on 01/05 at 6 h the simulated value is as low as -520 ‰ against -430 ‰ for the top inlet. These depletion events do not correspond to any parallel signal in the simulated meteorological data.

We note that some of the observed day-to-day variability is associated with the occurrence of snowfall events during the night, between 20 h and 8 h, which took place on 2013/12/19 and 2014/01/06 and interfere with the diurnal variability. This effect may be related to the impact of cloudiness, which reduces the long wave radiative loss at night and warms the surface (Van den Broeke et al., 2006). Model mismatches for humidity and temperature may also be related with

incorrect simulation of the cloud cover (e.g. during 12/19 or 01/10).

We finally focus on deuterium excess. The observed mean value at 3 m is ∼30 ‰, while ECHAM5-wiso produces a mean value of ∼26 ‰, and LMDZiso produces a stable and low value of ∼9 ‰. The interdiurnal standard deviation is 5 ‰ for ECHAM5-wiso and 9 ‰ for our observations. ECHAM5-wiso therefore simulates deuterium excess values close to the observations within these uncertainties.

Surprisingly, some of the air masses simulated by ECHAM5-wiso have a correct deuterium excess variability (e.g. on 01/05 and 01/13), despite the fact that the model is not able to simulate correctly the magnitude of the $\delta D$ depletion during these days, as reported above.

Table 5 presents a synthesis of the linear relationships between the deuterium versus different parameters (temperature, humidity mixing ratio, $\delta^{18}O$ and d-excess) for the observed and simulated

values on two different time scales (day-to-day and diurnal). The left column corresponds to the 29 daily mean values we have calculated before. Only $\delta D$ vs $\delta^{18}O$ presents a strong linear relationship on this time scale, because the day to day variability of other parameters (mixing ratio, temperature) is much weaker than their diurnal variations (Sect. 4.2). The strong linear relationship on the diurnal scale highlights the importance of the local processes, which will be investigated through a box

model in Sect. 4.3.

## 4.2 Observed and simulated diurnal cycles

We present in Fig. 3 the diurnal variation of six parameters stacked over 18 days (labeled as 'selected diurnal cycles' in Fig. 2) with a 1 h resolution. The reader should notice that the wind components from the models are simulated at 10 m, whereas weather observations are located at 2 m.

We have chosen 18 days showing a strong diurnal variation in the humidity mixing ratio and the isotopes for the purpose of stacking them: for each of these days and for each parameter, we have removed the daily mean value to get the anomalies. Then from these anomalies we have calculated for each hour (from 1 to 24, hours UTC) the average of the 18 values and its associated standard deviation. For visualization purposes we have calculated the average of the 24 standard deviations,





called hereafter MSD for 'Mean Standard Deviation' (each error bar in Fig. 3 represents ±1 MSD). The saturated mixing ratio has been shifted to preserve $(q_{sat} - q)$ and allow a meaningful comparison with the humidity mixing ratio anomalies. The averaged temperature (averaged humidity mixing ratio at 3 m) over the 18 selected days is -23.6±1.9 °C (1130±170 ppmv), which is close to the mean values over the campaign (-23 °C, 1100 ppmv) and indicates that these days constitute a representa-

tive sample of the campaign. We will also frequently refer to the measurements performed by Van As et al. (2005) at Kohnen from 2002/01/08 to 2002/02/09 in temperature and specific humidity at two different heights : surface and 1 m.

The daily variability of the measured wind direction can hardly be interpreted, because the large MSD of 66 ° exceeds the amplitude of the variations (61 °). The mean wind direction over the

18 selected days is 80±50 ° (the uncertainty is the standard deviation on the average), consistent with the slope direction of the terrain (61 °) in the presence of katabatic winds. Concerning the wind speed, the mean value is 4.4±1.6 m.s$^{-1}$ over the selected days, with a sharp decrease down to 3.4 m.s$^{-1}$ observed from 12 h to 20 h (UTC time). Both models simulate a mean wind direction close to 61 ° (60±40 ° for ECHAM5-wiso and 70±50 ° for LMDZiso), however the simulated wind

speeds have a lower diurnal variability, possibly due to the height in the simulation (10 m) compared to observation. However, the daily amplitude is 1.1 m.s$^{-1}$ for ECHAM5-wiso and 0.6 m.s$^{-1}$ for LMDZiso, whereas Van As et al. (2005) observed at the same height (10 m) a variability higher than 2 m.s$^{-1}$. The underestimation might be due to the horizontal resolution, which is too coarse to represent properly the katabatic winds, especially in LMDZiso.

We focus on the amplitude of the diurnal cycles, summarized in Table 4. The diurnal amplitude of observed surface air temperature is 10.0 °C (within a MSD of 1.0 °C). Van As et al. (2005) found an amplitude of ~ 14 °C at the surface and ~11 °C at 1 m, which is consistent with our observations. For the measurements from the 3 m inlet, we find a diurnal amplitude for $q$ of 930 ppmv and for $\delta D$ of 36 ‰. The magnitude of mean daily $\delta D$ and temperature variations is similar to that observed

in Greenland at NEEM (~35 ‰ for $\delta D$ and ~10 °C), despite much larger humidity variations in Greenland with ~ 2300 ppmv (Steen-Larsen et al., 2013). Both models strongly underestimate the diurnal amplitude of the temperature and the humidity mixing ratio variations at Kohnen. Surprisingly ECHAM5-wiso manages to simulate the right magnitude of the diurnal deuterium variability (32 ‰), however with a large MSD of 15 ‰ (Table 4).

We now compare the amplitudes at 0.2 and 3 m for the humidity mixing ratio and the $\delta D$. The absolute difference between the amplitudes measured from the bottom and the top inlets is 80 ppmv for $q$ (20 % of $\Delta q$) and 4 ‰ for $\delta D$. The average of the standard deviations associated with the minima and maxima from both inlets give ~125 ppmv for $q$ and ~6 ‰ for $\delta D$, and we cannot conclude that the amplitude is decreasing significantly with height. However, the strong diurnal cycle

in specific humidity, in phase with temperature, is unlikely to be caused by synoptic variability, and





we hypothesize that the diurnal variations in q are in fact due to evaporation and condensation. This hypothesis will be tested using a simple box model in section 4.3.

In the observations, diurnal air temperature variations occur in phase with $\delta D$ and humidity mixing ratio measured at 0.2 m, with minima at 2 h and maxima at 16 h. While $\delta D$ observations from 0.2

and 3 m seem synchronous within uncertainties, the humidity mixing ratio measured from the top inlet presents a clear delay of 1 hour. The same delay was observed by Van As et al. (2005) from the surface to 1 m in both temperature and humidity mixing ratio. Their observations showed minima and maxima for $T$ and $q$ at ~3 h (time UTC) and ~16 h at 1 m against ~2 h and ~15 h at the surface. The diurnal observations from NEEM (Steen-Larsen et al., 2013) were also in phase with $T$, $q$ and

$\delta D$ at ~1 m, however humidity mixing ratio and $\delta D$ measured on a much higher tower (~13 m) showed a delay of only ~1 h with the observations at 1 m. In the simulations from ECHAM5-wiso and LMDZiso, temperature diurnal variations occur approximately synchronous with the observations at Kohnen. However, simulated humidity mixing ratio and deuterium variations are delayed by ~3 hours compared to the observations from the 0.2 and 3 m inlets. This could be explained by

the fact that the temperature at 2 m is driven by the surface radiative budget while the timing of changes in humidity/isotopes may reflect boundary layer dynamics which are less accurately simulated. These simulated values are representative of the height of the first model level, which is also not expected to correctly estimate the observed values measured close to the surface.

The d-excess values also depict a diurnal cycle anti-correlated to $\delta D$, with an amplitude for the

0.2 m inlet (3 m inlet) of 21 ‰ (15 ‰) but associated with a large MSD: 7 ‰ (6 ‰).This anti-correlation is expected from the d-excess linear definition: at very low temperature, d-excess is influenced by distillation, and increases as $\delta D$ decreases. The interdiurnal variability is much weaker from 9 h to 14 h (~5 ‰ for the bottom inlet), but the standard deviations associated to the peak from midnight to 4 h are higher (~9 ‰ for the bottom inlet). ECHAM5-wiso underestimates the diurnal

amplitude variability and LMDZiso fails to simulate any diurnal variability in the d-excess, possibly because it simulates higher temperatures, where the distillation effect on d-excess is weaker.

As previously reported for daily mean values (Table 5, Sect. 4.1), close linear relationships are observed between $q$, $\delta D$ and d-excess for hourly mean values, highlighting the importance of local fluxes. These relationships are better simulated for the diurnal cycle, but ECHAM5-wiso tends to

overestimate the associated slopes. LMDZiso strongly overestimates the slope of $\delta D$-$\delta^{18}O$, possibly due to its warm bias. ECHAM5-wiso is able to capture the diurnal anti-correlation of d-excess and $\delta D$. The slope calculated on the hourly scale for $\delta D$ vs $\delta^{18}O$ has a value of 5.99±0.12, whereas Steen-Larsen et al. (2013) calculated at NEEM a slope of 6.47±0.07, in a warmer and more humid air (~3000 ppmv against ~1200 ppmv at Kohnen).





### 4.3 Air-snow exchanges

In order to document the water exchanges between the surface snow and the overlying vapor, we sampled the snow surface hourly for 35 hours on a clear day (Section 3.5). We sampled the top 2-5 mm of three snow patches every hours, with 5 juxtaposed replica for each patch (15 samples per hour). Each hourly data point is the average of the five snow samples, and its standard deviation is 0.03 ‰ for $\delta^{18}O$ and 0.2 ‰ for $\delta D$. We identify a clear diurnal cycle in the snow with a significant peak-to-peak amplitude of $\sim 3$ ‰ for $\delta D$ and $\sim 0.4$ ‰ for $\delta^{18}O$, in phase with the diurnal cycle in the air (Fig. 4). Unfortunately, the cooling phase during the night of 2014/01/09 is restricted compared to the usual strong decreases in temperature or humidity shown in Fig. 3 because the presence of a cloud cover. This meteorological event is likely to have impacted $T$, $q$ and $\delta D$ during the cooling phase.

We notice that the vapor is close to or at saturation by looking at the saturated mixing-ratio calculated in Fig. 4. We can therefore expect condensation to occur during the night (and sublimation during the day as the diurnal cycle is observed to be approximately symmetrical) and an isotopic exchange between the lower atmospheric water vapor and the surface snow. This raises the question, if and by which magnitude condensation and sublimation processes might affect the surface snow and lower the water vapor isotopic composition.

In order to address this question, we set up a simple box model as a closed system containing two interacting and homogeneous reservoirs. Figure 5 depicts the schematics of this model, and introduces our notations. As this system is closed, the variation of moisture in the air (and its isotopic composition) is only due to condensation/sublimation during the cooling/warming phase and we have mass conservation of the water molecules: $\forall t,\ \mathrm{m}_t^v + \mathrm{m}_t^s = \mathrm{m}_0^v + \mathrm{m}_0^s$ with $t = 0$ the start of the cooling phase and $v$ and $s$ indices representing vapor and snow, respectively. Our simulation is based on the values from the mean stack of the 18 diurnal cycles instead of the specific day corresponding to the snow sampling because of the unusually high temperature and humidity during the night of 2014/01/09. We split our analysis into two parts : the cooling phase (from $\sim 17$ h to $\sim 2$ h UTC) and the warming phase (from $\sim 3$ h to $\sim 15$ h UTC).

#### 4.3.1 Cooling phase

The deposition of the condensate on the snow surface during the cooling phase is expected to reach a maximum height of $\zeta^{\mathrm{max}} \sim 0.1$ mm (calculation detailed in Fig. 5). As the depth of our surface snow samples is $\sim 2 - 5$ mm, we mix the condensate with the snow reservoir. From $t$ to $t + 1$, an amount $(\mathrm{m}_t^v - \mathrm{m}_{t+1}^v)$ condensates and the isotopic ratio of the condensate in equilibrium with the vapor is $\alpha_t \mathrm{R}_t^v$ with R the isotopic ratio of the heavy isotope and $\alpha_t$ the associated fractionation coefficient with respect to ice calculated with the air temperature measured at 2 m at time $t$. Assuming an immediate removal of the condensate from the air reservoir and an immediate mixing with the snow





reservoir, we obtain for the isotopic composition of the vapor $\delta_t^v$ and the isotopic composition of the snow $\delta_t^s$:

$$\delta_{t+1}^v + 1000 = A_t^v(\delta_t^v + 1000) \tag{2}$$

$$\delta_{t+1}^s + 1000 = B_t^s(\delta_t^s + 1000) + B_t^v(\delta_t^v + 1000) \tag{3}$$

with

$$A_t^v = \frac{q_t}{q_{t+1}} - \alpha_t\left(\frac{q_t}{q_{t+1}} - 1\right)$$

$$B_t^s = \frac{\rho_s h_0 + \rho_d H_0(q_0 - q_t)}{\rho_s h_0 + \rho_d H_0(q_0 - q_{t+1})}$$

$$B_t^v = \alpha_t \frac{\rho_d H_0(q_t - q_{t+1})}{\rho_s h_0 + \rho_d H_0(q_0 - q_{t+1})}$$

We will consider the equilibrium case (relative humidity RH set to 1) but also the supersaturated case (RH = 1.1) by replacing $\alpha_t$ with the equivalent fractionation coefficient (Jouzel and Merlivat, 1984), which takes into account the kinetic effects in a supersaturated environment. The required input parameters for simulating $\delta_t^v$ are $q_t$, $\delta_0^v$, $\alpha_t$ and RH (set to 1 or 1.1). The required input param-

eters for simulating $\delta_t^s$ are $q_t$, $\delta_t^v$ (measured, not simulated), $\alpha_t$, $h_0$, $H_0$, RH, $\rho_s$, $\rho_d$ and $\delta_0^s$.

Figure 6 presents the simulation of $\delta D_t^v$ during the cooling phase, based on equations (2) and the isotopic variation of the condensate (Eq. (3) with $h_0 = 0$ mm), which does not depend on $H_0$, $\rho_s$ or $\rho_d$. We have used fractionation coefficients with respect to ice given by Merlivat and Nief (1967) and Ellehoj et al. (2013).

The amplitude of the simulated isotopic composition of the vapor is for each case three times larger than observed. The box model is closed, so any change in the vapor is forced to condensate in order to keep the mass conservation equation. In reality, there is wind advection but also possible exchanges with the free troposphere, which could partly contribute to the decrease of humidity during the cooling phase in an open system instead of a pure condensation process in a closed system.

Nevertheless, our simplistic approach leads to the conclusion that about 40% of the diurnal vapor mixing ratio variation is sufficient to simulate the right order of magnitude of isotopic variations, based on equilibrium fractionation. This is consistent with the results of ECHAM5-wiso: while this atmospheric model underestimates the diurnal variability of humidity (by 40%), it does correctly capture the diurnal variability of deuterium.

The isotopic variation of the condensate is ~6 ‰ and decreases in phase with the vapor. We define $(\delta D_0^s)_{eq}$ as the deuterium value of the condensate at equilibrium with the initial vapor. The value of





$(\delta D_0^s)_{\text{eq}}$ is -295 ‰ using the fractionation coefficient from Merlivat and Nief (1967). This result is consistent with the mean value of the isotopic composition of the three snow patches (Table 6).

Figure 7 presents the simulation of $\delta D_t^s$ during the cooling phase, based on equations (3). We have

used $\rho_d = 0.95$ kg.m$^{-3}$, calculated from the surface pressure, temperature and relative humidity measured at Kohnen, and $\rho_s = 340$ kg.m$^{-3}$, calculated from 100 daily snow samples collected at Kohnen during the period of air measurement. The polar boundary layer height is expected to have a value between 50 and 100 m, and the snow reservoir a depth between 2 and 5 mm due to the uncertainty of the snow surface sampling. We have chosen two initial isotopic compositions of the

snow surface as two distinct cases : (i) $\delta D_0^s = -290‰$, above $(\delta D_0^s)_{\text{eq}}$ and (ii) $\delta D_0^s = -310‰$, below $(\delta D_0^s)_{\text{eq}}$.

In the first case, the mixing between the condensate and the snow surface will tend toward the equilibrium in a decreasing trend and in the second case the trend is predicted as increasing. This is due to the difference between the isotopic composition of the condensate and the snow surface,

negative or positive at a given time $t$. The response of the model shown in Fig. 7 depends strongly on parameters that are not well constrained. These parameters are the box sizes (a snow reservoir with a depth above 1 cm will keep a constant isotopic composition), the fractionation coefficients (disagreement between Merlivat and Nief (1967) and Ellehoj et al. (2013)) or the value of $\delta D_0^v$, which could be measured with an accuracy of 11‰ (see Table 2) only. However, we are able to

conclude that the condensation of water vapor is the likely cause of the observed changes in the isotopic composition of the top 2 mm of the snow surface.

### 4.3.2   Warming phase

From $t$ to $t+1$, an amount $(m_{t+1}^v - m_t^v)$ sublimates and the isotopic ratio $R_t^{\text{sub}}$ of the sublimate will be tested under three different hypotheses: (i) no fractionation occurs and $R_t^{\text{sub}} = R_t^s$ (ii) the

sublimate is formed in equilibrium with the snow and $R_t^{\text{sub}} = R_t^s/\alpha_t$ (iii) the kinetic effect due to subsaturation is taken into account and a thin layer of liquid water above the snow with the same isotopic composition is considered. Following Merlivat and Jouzel (1979) we have in case (iii):

$$R_t^{\text{sub}} = \frac{1-k}{1-\text{RH}}\left(\frac{R_t^s}{\alpha_t} - \text{RH} \times R_t^v\right)$$

With RH the relative humidity set equal to 0.9, and k the kinetic fractionation factor given by

$k_{\delta^{18}O} = 6.2‰$ and $k_{\delta D} = 5.5‰$. We present the equations for the vapor and the snow surface for the case (iii) only, noticing that cases (i) and (ii) are mathematically obtained from case (iii).

Assuming an immediate removal of the sublimate from the snow reservoir and an immediate mixing with the molecules contained in the air reservoir, we have for the isotopic composition of the vapor:

$\delta_{t+1}^v + 1000 = E_t^v(\delta_t^v + 1000) + E_t^s(\delta_t^s + 1000)$          (4)




with

$$E_t^v = \frac{q_t}{q_{t+1}} - RH \times \frac{1-k}{1-RH}\left(1 - \frac{q_t}{q_{t+1}}\right)$$

$$E_t^s = \frac{1}{\alpha_t}\frac{1-k}{1-RH}\left(1 - \frac{q_t}{q_{t+1}}\right)$$

The equations for the isotopic composition of the snow are:

$$\delta_{t+1}^s + 1000 = F_t^s(\delta_t^s + 1000) + F_t^v(\delta_t^v + 1000) \tag{5}$$

with

$$F_t^s = \frac{\rho_s h_0 + \rho_d H_0\left(q_0 - q_t - \frac{1}{\alpha_t}\frac{1-k}{1-RH}(q_{t+1} - q_t)\right)}{\rho_s h_0 + \rho_d H_0(q_0 - q_{t+1})}$$

$$F_t^v = RH \times \frac{1-k}{1-RH} \times \frac{\rho_d H_0(q_{t+1} - q_t)}{\rho_s h_0 + \rho_d H_0(q_0 - q_{t+1})}$$

We notice that $\delta_{t+1}^s = \delta_t^s$ when no fractionation occurs as it is assumed in ECHAM5-wiso and
LMDZ-iso. Steen-Larsen et al. (2011) and Landais et al. (2012) showed based on Greenland data
that on average the snow surface isotopes and the water vapor isotopes are in equilibrium. They have
calculated that the value of the equilibrium factor is between the fractionation coefficient $\alpha^{ice}$ with
respect to ice (Merlivat and Nief, 1967; Ellehoj et al., 2013) and the fractionation coefficient $\alpha^{water}$
with respect to water (Majoube, 1971). We therefore present the results simulated with the different
fractionation coefficients (Merlivat and Nief, 1967; Ellehoj et al., 2013; Majoube, 1971) to get an
estimate of the uncertainties.

Figure 8 displays the measurements and simulations performed for the isotopic composition of
the deuterium and d-excess of the vapor during the warming phase. When no fractionation occurs
during sublimation, the simulated variation of the deuterium is two times higher than observed. If we
were sublimating a block of solid ice, it would be conceivable that only the very surface atoms would
be able to sublimate, and the system would not fractionate. However, in the presence of very porous
snow, there are a very large numbers of water molecules participating in the snow-air interface, and
it is conceivable that snow would behave more like a liquid than like a solid in this respect, and
fractionate. We tested for the presence of fractionation by running the box model with a variety of
available fractionation factors for air over ice and water (Fig. 8), and find that using $\alpha^{ice}$ the model
underestimates the variations in $\delta D$. The true fractionation factor at sublimation is probably lower
than $\alpha^{ice}$, but the crude nature of our model prevents us from quantifying it precisely.

Figure 9 presents the simulated $\delta D$ of the surface snow during the warming phase. A difference
between equilibrium and subsaturation has to be noticed in Eq. (5) due to the coefficient $F_t^v$. If
$F_t^v = 0$ (equilibrium), there is no influence of the vapor on the isotopic composition of the surface
snow and $\delta_0^s$ will not not have a significant impact on $\delta_t^s$: any patch of snow will share the same





isotopic variation whatever its initial isotopic composition is. If $F_t^v \neq 0$ (subsaturation), the isotopic

composition of the snow is affected by the isotopic composition of the vapor, hence a different trend of $\delta_t^s$ depending on $(\delta_0^v)_{eq}$ and $\delta_0^s$ as shown previously for the cooling phase (Fig. 7).

We focus here only on $\delta D_0^s$ = -320 ‰, which is the average of the 100 daily snow samples collected at Kohnen over the measurement period. Our data from the three snow patches consistently depict a positive trend during the warming phase, with an amplitude between 3 and 7 ‰ for $\Delta \delta D$ (Fig. 9).

Simulations with different values of $H_0$, $h_0$ and $\alpha_t$ share the same positive trend with a peak-to-peak amplitude between 1 and 8 ‰, which is of the order of magnitude as the observations. We notice that the uncertainties related to the reservoirs heights have greater impacts on the simulated snow surface isotopic values than the different fractionation coefficients. As a result, we are not able to constrain the fractionation factor at sublimation, but we observe that since the surface snow isotopic

composition is changing, the sublimation process must be associated with an isotopic fractionation.

## 5   Conclusions

Continuous measurements of temperature, humidity mixing ratio and water vapor isotopes were performed during summer 2012/2013 at Kohnen station in East-Antarctica. These data highlight a strong diurnal cycle, in contrast with rather stable day-to-day mean levels over 1 month of obser-

vations. During our monitoring period, the surface vapor isotopic composition was therefore more driven by local processes than by synoptic changes. This motivated us to investigate the diurnal isotopic response of the upper thin layer of snow surface to the atmospheric variations. A continuous hourly sampling over 35 hours of the first ∼2-5 mm of the snow surface of three different snow patches reveals a significant variability in both $\delta D$ and $\delta^{18}O$ during a period without snowfall events.

As these variations in the surface snow isotopic composition follow the diurnal trend in the air, this striking result confirms the observations of Steen-Larsen et al. (2013) at NEEM who also observed parallel variations between the snow surface isotopic composition and lower atmosphere isotopic composition. These observations were however with a higher variation on a day-to-day scale. In their case, they reported larger variations in the parallel variations in the isotopic composition of

both surface snow (5 mm) and vapor, reaching 10 ‰ over 5 days.

Two important consequences can be inferred from the snow sample diurnal observations: (1) post-depositional processes have a significant impact on the isotopic composition of the snow surface and (2) the sublimation process is fractionating. These two points are not included in classical isotopic theory and therefore not implemented in atmospheric models.

In order to determine the contributions of condensation and sublimation to the isotopic variations of the vapor and surface snow, we developed a simple model describing the isotopic exchange between two reservoirs contained in a closed system: a water vapor column and a thin snow surface layer. We find that the observed isotopic variations in the water vapor phase ($\delta D^v$) are about half





of what simple condensation/evaporation equilibrium would dictate. It is likely due to advection,

exchanges with the free troposphere and variations in the boundary layer height. Additionally, for our observed isotopic composition of surface (2-5mm) snow changes during both the warming and the cooling phase (peak-to-peak variation of $\sim 3$ ‰ for $\delta D$), our crude model is able to reproduce these observations, although the model results depend strongly on the size of the reservoir chosen. For instance, if we were to sample snow over 1 cm in thickness, the diurnal cycle in $\delta D^s$ would no

longer be measurable. We do observe an increase in $\delta D^s$ during the sublimation process, which indicates that water isotopes fractionate during sublimation, except if the wind removes layers of surface snow. However, the uncertainties in the model geometry, and in air advection prevent us from being able to determine the fractionation coefficient. Our water vapor isotope data suggest however that it is smaller than $\alpha^{\text{ice}}$.

The day-to-day variations in water vapor isotopic composition have a much smaller amplitude than the diurnal cycle, partly because no large synoptic event was recorded during our monitoring period. Expanding the temporal framework of such monitoring is a pre-requisite in order to better understand the importance of horizontal advection, and to evaluate the processes at play during the winter season. Our observations show the possible importance of surface snow - surface vapor exchanges

for the isotopic composition recorded in ice cores. This stresses the potential of isotopic monitoring of snow-air interactions for the study of fractionation processes during water phase change, but also underscore the importance of improvements in analytical accuracy under low humidity conditions. This constitutes an experimental challenge for future works.

*Acknowledgements.* This first paper is dedicated to Dominique Hirondel, former teacher of history of sciences,

for being a role model for scientific curiosity and fascination toward reality. The scientific campaign would not have been possible without the efficient work of the AWI logistic staff. Special thanks also to Carleen Reijmer for providing the data of AWS9, to Michael Schaefer, Gert Koenig-Langlo and Bernd Loose for their contribution to the meteorological observations and to Max Berkelhammer for his valuable contribution to the final version of the manuscript. LMDZiso simulations were performed on the Ada supercomputer at IDRIS

under GENCI project 0292.





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





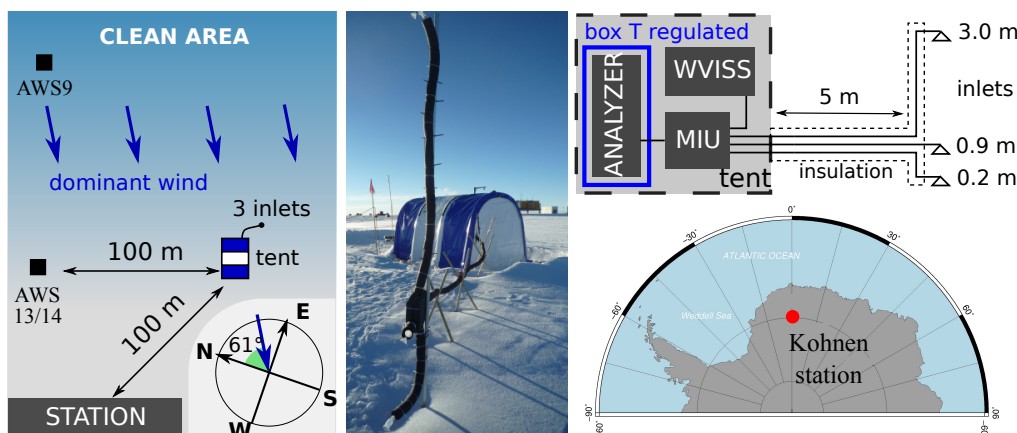

**Figure 1.** Location of Kohnen station in Antarctica (bottom right panel). Location of the two Automatic Weather Stations (AWS, left panel) and location of the measurement tent connected to the three inlets (central panel). The right schematic presents the set-up with the three inlets, the multiport (MIU), the Water Vapor Isotope Standard Source (WVISS) and the analyzer measuring the humidity mixing ratio $q$ and isotopes $\delta D$ and $\delta^{18}O$ in the vapor.





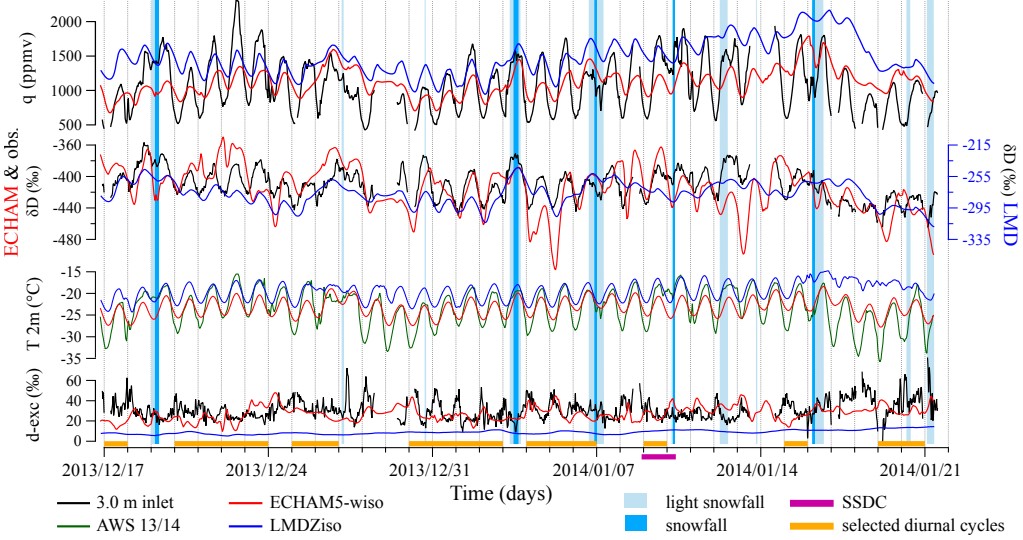

**Figure 2.** Hourly observed (black and green) and simulated (red and blue) humidity mixing ratio ($q$), deuterium ($\delta D$), air temperature at 2 m ($T$ 2m, measured with the AWS from the 2013/2014 season) and d-excess (d-exc) at Kohnen station 2013/2014. LMDZiso has its own axis with respect to the deuterium. Hourly observed precipitation events are labelled light snowfall and snowfall (light and dark cyan). The Snow Surface Diurnal Cycle (SSDC) experiment period is indicated in purple, and the 18 selected days for the diurnal cycles study in orange.



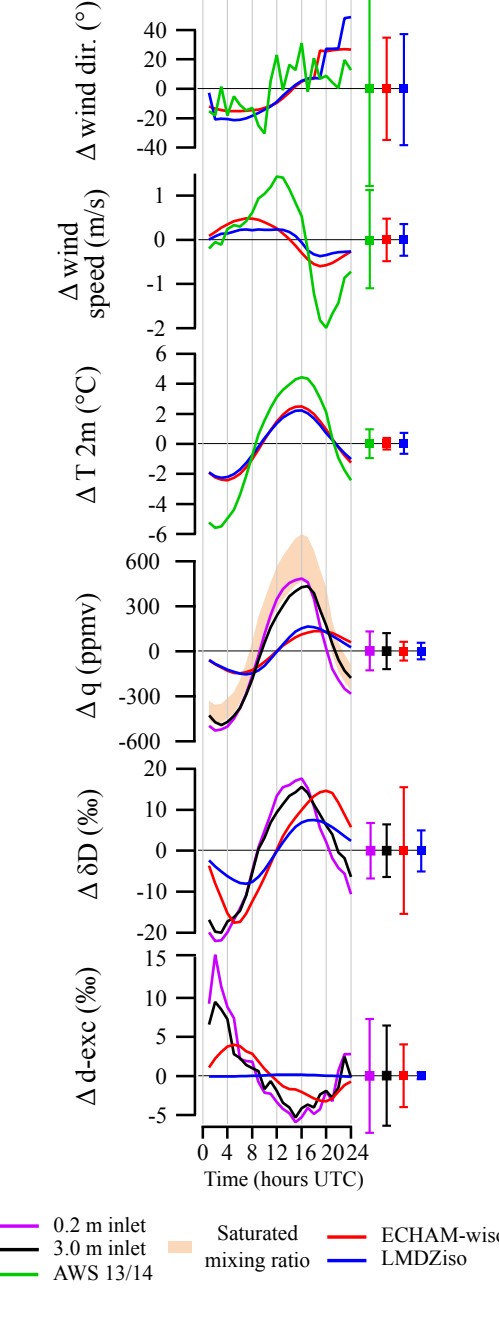

**Figure 3.** Stack of 18 diurnal cycles of the wind direction and wind speed (simulated at 10 m for the models, measured at 2 m for the observations), air temperature at 2 m (measured with the AWS from the 2013/2014 season), humidity mixing ratio $q$, saturated mixing ratio range calculated with the surface pressure and temperature at 2 m $\pm 1$ °C, deuterium $\delta D$ and d-excess. Error bars represent $\pm 1$ mean standard deviation (MSD): average of the 24 standard deviations associated with the hourly mean values.

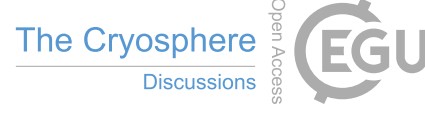



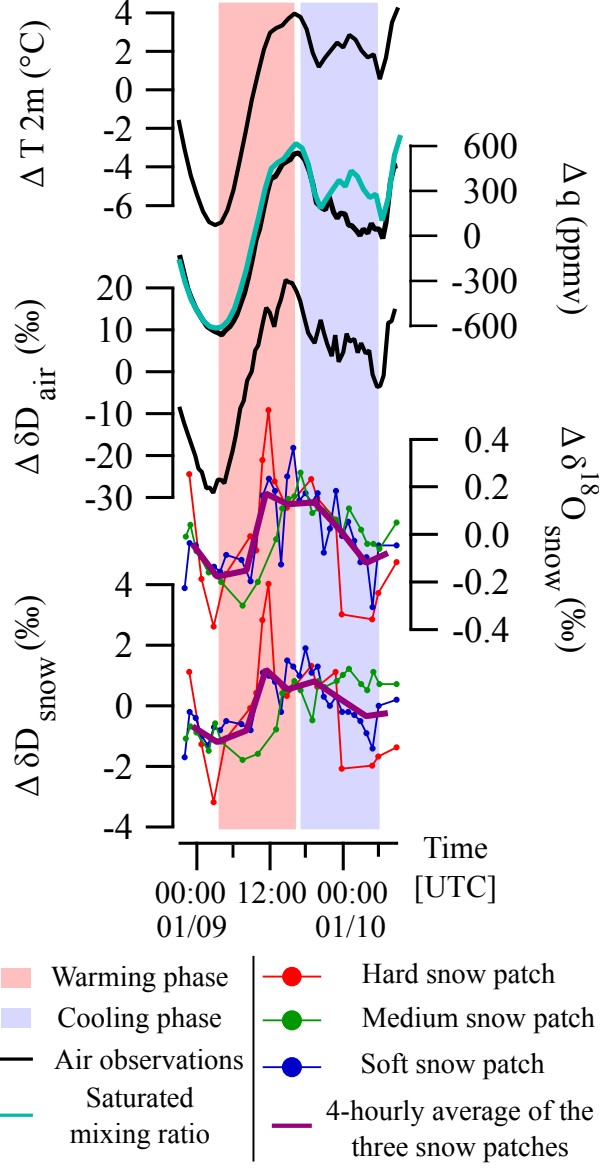

**Figure 4.** Observations performed during the Snow Surface Diurnal Cycle (SSDC) experiment from the 2014/01/09 to the 2014/01/10. The three different snow patches are labelled 'Hard', 'Medium' and 'Soft' from their texture. Each hourly data point is the average of the measurements made on 5 snow surface samples per patch.





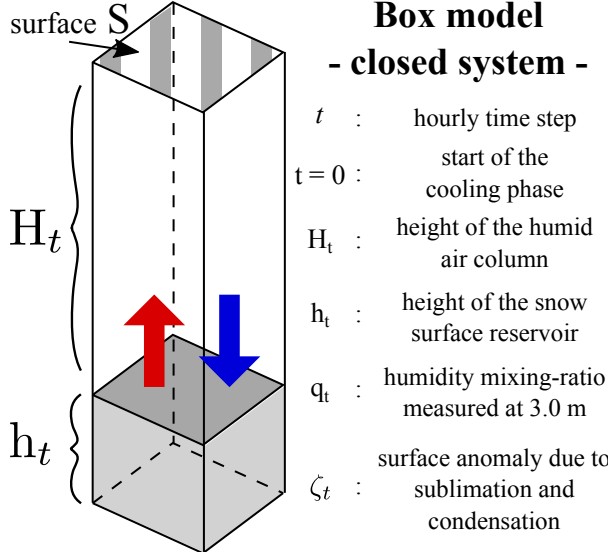

# Box model
## - closed system -

$t$ : hourly time step

$t = 0$ : start of the cooling phase

$H_t$ : height of the humid air column

$h_t$ : height of the snow surface reservoir

$q_t$ : humidity mixing-ratio measured at 3.0 m

$\zeta_t$ : surface anomaly due to sublimation and condensation

$\rho_d , \rho_s$ : dry air density and snow density

$m_t^v, m_t^s$ : water vapor mass and snow mass

$$h_t = h_0 + \zeta_t \qquad m_t^s = \rho_s h_t S$$
$$H_t = H_0 - \zeta_t \qquad m_t^v = q_t \rho_d H_t S$$

**Snow accumulation during condensation:**

$$\zeta_{t+1} \approx \zeta_t + \frac{\rho_d}{\rho_s} H_0 (q_t - q_{t+1})$$

**Order of magnitude of the parameters:**

$$\rho_d = 0.95 \text{ kg.m}^{-3} \qquad h_0 \approx 2\text{-}5 \times 10^{-3} \text{ m}$$
$$\rho_s = 300\text{-}380 \text{ kg.m}^{-3} \qquad q_t \in 5\text{-}9 \times 10^{-4} \text{ kg/kg}$$
$$H_0 \approx 50\text{-}100 \text{ m} \qquad \zeta^{max} \approx 1.1\text{-}1.4 \times 10^{-4} \text{ m}$$

**Figure 5.** Schematic of the box model and description of the input parameters (isotopes excluded). $\zeta^{max}$ is the maximum value of the anomaly at the end of the cooling phase, calculated with two extrem values of ($H_0, \rho_s$). The height of the SSDC snow samples gives an estimation of $h_0$. $\rho_d$ and $\rho_d$ have been calculated from 100 snow samples per day, surface pressure, temperature and relative humidity observed at Kohnen from 2013/12 to 2014/01.





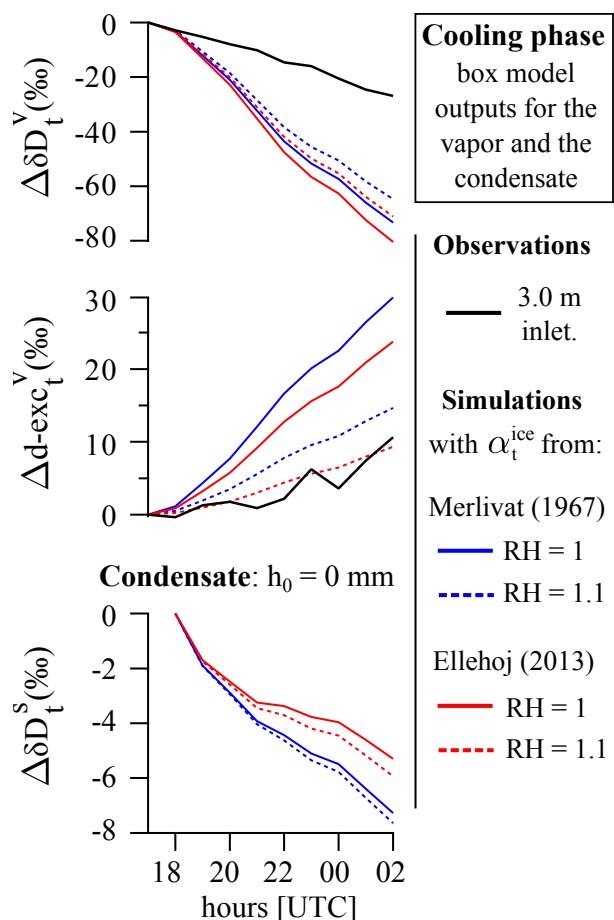

**Figure 6.** Measured and simulated (box model) deuterium anomalies of the vapor and simulated deuterium anomaly of the condensate ($h_0$=0 mm) during the cooling phase, from 17 h to 2 h UTC. Observations come from the stack of the 18 diurnal cycles. Equivalent fractionation coefficients have been calculated from Jouzel and Merlivat (1984) for RH=1.1.





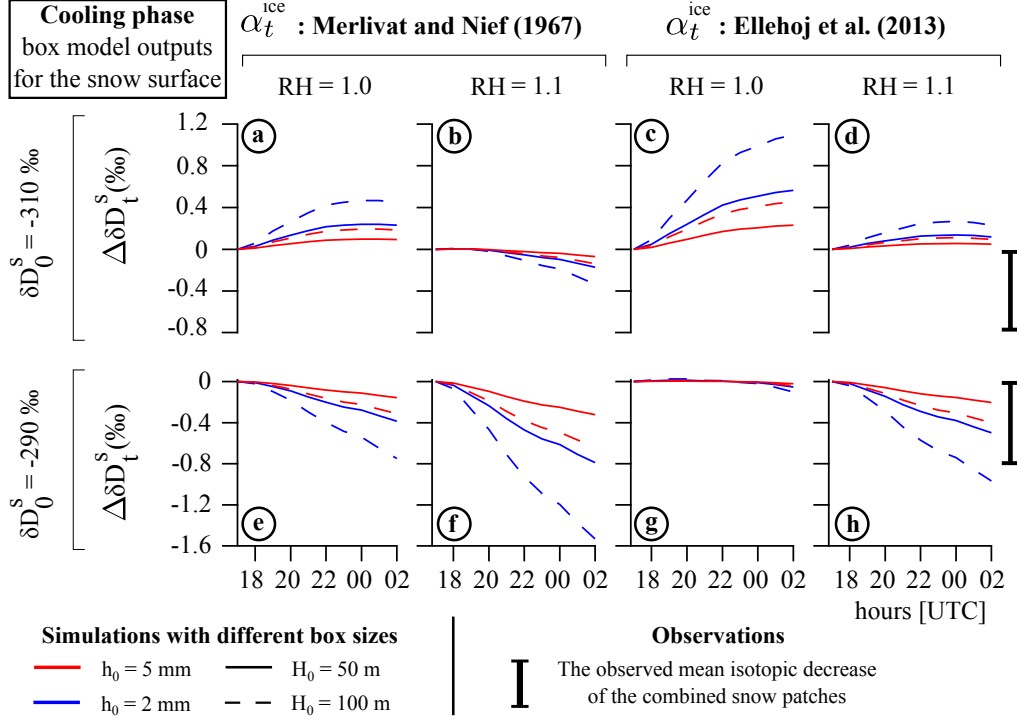

**Figure 7.** Simulated (box model) deuterium anomalies of the snow surface during the cooling phase, with $h_0$, $H_0$, RH, $\alpha_t$ and $\delta D_0^s$ as varying inputs parameters. Fractionation coefficients at equilibrium come from Merlivat and Nief (1967) and Ellehoj et al. (2013)., calculated with $T$ at 2 m. Equivalent fractionation coefficients have been calculated from Jouzel and Merlivat (1984) for RH = 1.1.



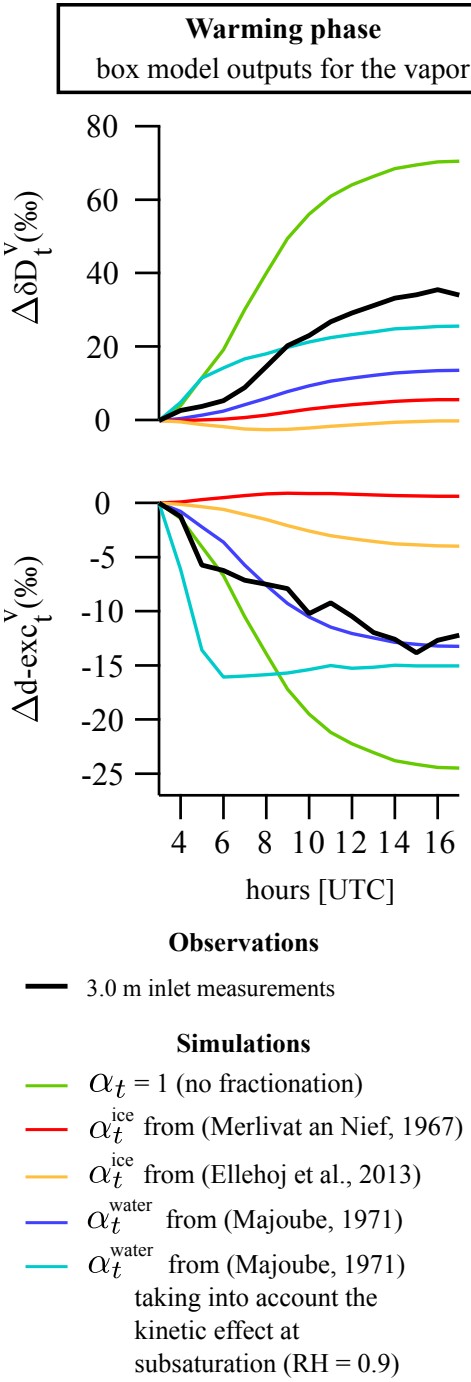

**Figure 8.** Measured and simulated (box model) deuterium and d-excess anomalies of the vapor during the warming phase. Kinetic effects occuring at subsaturation (RH = 0.9) have been calculated following (Merlivat et al., 1979) in a smooth regime.



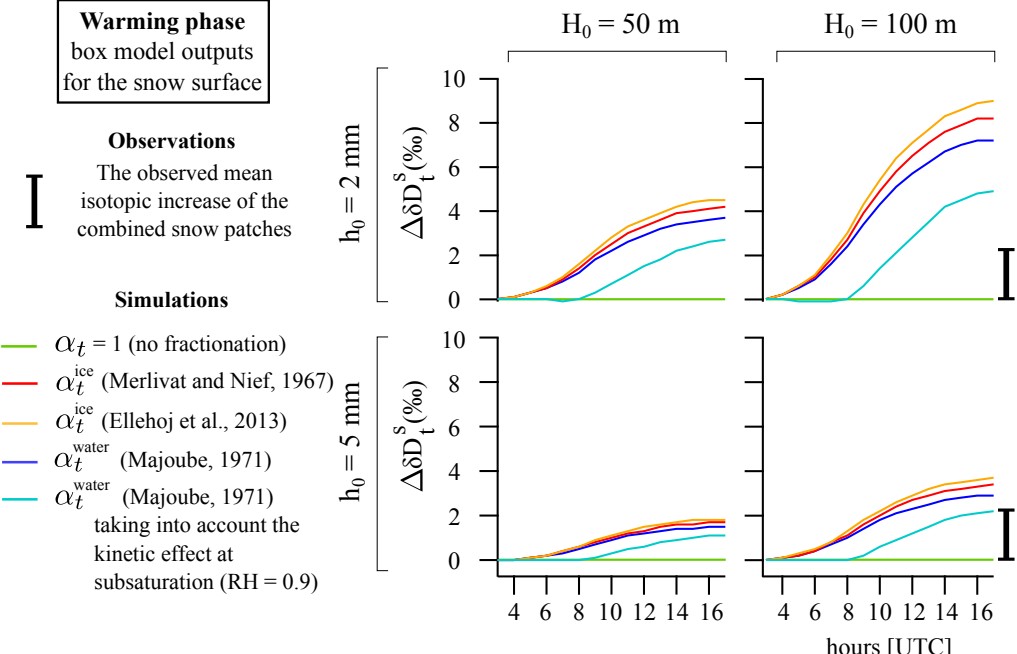

**Figure 9.** Simulated (box model) deuterium anomalies of the snow surface during the warming phase, with $h_0$, $H_0$ and the fractionation coefficients as varying inputs parameters. The initial isotopic composition of the snow surface has been taken as the average of the 100 daily snow samples collected at Kohnen over the period of measurements, $\delta D_0^s$ = -320 ‰.





| day of calibration | δD (standards) vs δD (measured) | $\delta^{18}O$ (standards) vs $\delta^{18}O$ (measured) | standards used |
|---|---|---|---|
| 2013/12/06 | 1.45 ± 0.02 | 1.03 ± 0.07 | NZE, JASE, TD1 |
| 2013/12/15 | 1.40 ± 0.03 | 0.93 ± 0.05 | DML, TD1, NZE, JASE |
| 2014/01/08 | 1.38 ± 0.01 | 1.01 ± 0.03 | NZE, OC3, TALOS, NEEM |
| 2014/01/22 | 1.47 ± 0.01 | 1.01 ± 0.01 | JASE, TD1, NZE, DML |

**Table 1.** Conversion slopes calculated from four VSMOW-SLAP calibrations with different standards. Data have been corrected with respect to humidity before calculating the slopes. Uncertainties represent 1 standard error on the slopes.

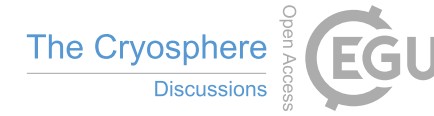

| | related to $\delta$D | | | related to $\delta^{18}$O | | |
|---|---|---|---|---|---|---|
| | avg | min | max | avg | min | max |
| $\alpha_*$ | 1.43 | / | / | 1.00 | / | / |
| $d\alpha_*$ | 0.02 | / | / | 0.03 | / | / |
| $\delta^*_{raw}$ | -552 | -586 | -519 | -84.3 | -94.6 | -77.7 |
| $d\delta^*_{raw}$ | 0.3 | 0.1 | 0.9 | 0.1 | 0.05 | 0.4 |
| $|\Gamma_*|$ | 4 | 0 | 9 | 1.7 | 0 | 4 |
| $d\Gamma_*$ | 2 | 0.1 | 5 | 0.4 | 0 | 2 |
| $d\mu_*$ | 1.4 | / | / | 0.7 | / | / |
| $A_*$ | 11 | 10 | 12 | 2.5 | 2.2 | 2.9 |
| $P_*$ | 3.0 | 1.4 | 7.3 | 0.9 | 0.7 | 2.0 |

**Table 2.** Order of magnitude of the parameters involved in the error propagation calculation for $\delta D$ and $\delta^{18}O$. 'avg', 'min' and 'max' are the mean, the minimum and the maximum value of the parameter over the campaign. 'A' stands for accuracy and 'P' for precision. Every parameter is in ‰, except $\alpha_*$ and $d\alpha_*$ which are dimensionless.





|            | q (ppmv)  | δD (‰)   | d-exc (‰)  | T 2m (°C)  |
|------------|-----------|----------|------------|------------|
| AWS 13/14  | /         | /        | /          | -23 ± 4    |
| 0.2 m inlet| 1000±200  | -413 ± 9 | 33 ± 11    | /          |
| 3.0 m inlet| 1100±200  | -409 ± 9 | 30 ± 9     | /          |
| ECHAM      | 1120±110  | -411±15  | 26 ± 5     | -23.1±1.7  |
| LMDZiso    | 1460±160  | -276±12  | 8.8 ± 0.6  | -19.7±1.8  |

**Table 3.** Mean values over 29 daily averages from the measurement period (days with more than 8 hours of data gap have been removed). Uncertainties represent ±1 standard deviation of the mean value. ECHAM stands for ECHAM5-wiso.





|  | $\Delta q$(ppmv) | $\Delta\delta D$(‰) | $\Delta$d-exc(‰) | $\Delta$T 2m(°C) |
|---|---|---|---|---|
| AWS 13/14 | / | / | / | 10.1 ± 1.0 |
| 0.2 m inlet | 1010 ± 130 | 40 ± 7 | 21 ± 7 | / |
| 3.0 m inlet | 930 ± 120 | 36 ± 6 | 15 ± 6 | / |
| ECHAM | 280 ± 60 | 32 ± 15 | 7 ± 4 | 4.9 ± 0.4 |
| LMDZiso | 320 ± 60 | 16 ± 5 | 0.3 ± 0.4 | 4.5 ± 0.7 |

**Table 4.** Peak-to-peak amplitude of the 18 selected diurnal cycles. Uncertainties represent ±1 mean standard deviation (average of the 24 standard deviations of the hourly mean values). ECHAM stands for ECHAM5-wiso.





| δD (‰) vs q (ppmv) | (1) daily mean values (n = 29) | (2) hourly mean values (n = 24) |
|---|---|---|
| 3.0 m inlet | α = 0.04 ± 0.01<br>r² = 0.31 | α = 0.037 ± 0.001<br>r² = 0.98 |
| ECHAM5-wiso | α = 0.03 ± 0.02<br>r² = 0.05 | α = 0.106 ± 0.003<br>r² = 0.99 |
| LMDZiso | α = 0.04 ± 0.01<br>r² = 0.44 | α = 0.049 ± 0.001<br>r² = 0.99 |
| **δD (‰) vs T 2m (°C)** | | |
| 3.0 m inlet | α = 3.1 ± 1.1<br>r² = 0.24 | α = 3.37 ± 0.11<br>r² = 0.98 |
| ECHAM5-wiso | α = 6 ± 4<br>r² = 0.10 | α = 4.7 ± 0.9<br>r² = 0.54 |
| LMDZiso | r² < 0.01 | α = 2.7 ± 0.5<br>r² = 0.55 |
| **δD (‰) vs δ¹⁸O (‰)** | | |
| 3.0 m inlet | α = 6.2 ± 0.3<br>r² = 0.94 | α = 5.99 ± 0.12<br>r² = 0.99 |
| ECHAM5-wiso | α = 6.64 ± 0.19<br>r² = 0.98 | α = 6.57 ± 0.04<br>r² = 0.99 |
| LMDZiso | α = 7.41 ± 0.18<br>r² = 0.98 | α = 8.06 ± 0.02<br>r² = 0.99 |
| **d-excess (‰) vs δD (‰)** | | |
| 3.0 m inlet | α = -0.21 ± 0.06<br>r² = 0.32 | α = -0.32 ± 0.03<br>r² = 0.86 |
| ECHAM5-wiso | α = -0.18 ± 0.03<br>r² = 0.50 | α = -0.22 ± 0.01<br>r² = 0.98 |
| LMDZiso | α = -0.06 ± 0.03<br>r² = 0.18 | α = 0.01 ± 0.01<br>r² = 0.23 |

**Table 5.** Slopes and determination coefficients calculated from two data sets : (1) the 29 daily mean values from the campaign, and (2) the 24 hourly mean values from the stack composed of 18 selected diurnal cycles. Uncertainties represent 1 standard error on the slopes.



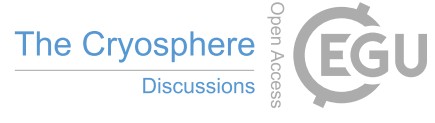

| | $\overline{\delta D}$ (‰) | $\overline{\delta^{18}O}$ (‰) |
|---|---|---|
| air observations | -407 | -54.4 |
| patch soft | -296 | -37.4 |
| patch medium | -301 | -37.0 |
| patch hard | -316 | -38.8 |

**Table 6.** Isotopic mean values over the SSDC period. The standard error on the mean value is below 1 ‰ for $\delta D$ and below 0.1 ‰ for $\delta^{18}O$.