# Peer review of "Isotopic exchange on the diurnal scale between near-surface snow and lower atmospheric water vapor at Kohnen station, East Antarctica"

_The Cryosphere, 2016_

## Referee Comment (RC1) · Anonymous Referee #1 · 18 Mar 2016

The paper deals with post-depositional processes that affect the isotopic composition of surface snow in Antarctica. An exact knowledge of these processes is important for a correct paleo climatic interpretation of ice cores. Recently, it was found in Greenland that the interaction between atmosphere and the snow surface between snowfall events influences the stable isotope ratio of the snow much more strongly than previously thought. The paper describes the first continuous measurements of stable isotope ratios of water vapor on the Antarctic plateau, which were combined with high-temporal resolution measurements of the isotopic composition of surface snow. It is a highly interesting paper with important results and certainly worth to be published in TC. It is mostly well written, the English is generally ok, but not free of small errors (tenses etc.)

[Figure]

and slightly awkward formulations, on which I won't always comment in detail since I believe that this is not the job of a reviewer (while professional companies charge a lot of money for it). This should be checked. The description of previous work could be a bit more elaborate. I do not have expertise in measuring with, and calibration of, a Picarro, so I cannot assess this part. Generally, it is described in detail and seems to have been done with highest diligence. I have a couple of small specific comments (see below). My recommendation is to publish the paper after minor revision.

Specific comments:

32: what is a meteorological time scale? Better use "synoptic" if that is what you mean. (e.g. diurnal variations are also meteorological)

45: to avoid any confusion, please define humidity mixing ratio

91ff: How do you define a precipitation event? It is not correct that precipitation occurs only in summer, and at Kohnen station, only about 20% of snowfall events is directly related to synoptic systems. (additionally, there is diamond dust precipitation, which is not so rare). I personally don't like reviewers, who always want their own papers quoted, but, of course, everybody knows their own papers best and, unfortunately, I do not know any other paper that investigates Kohnen precipitation but Schlosser et al. (2010), which gives some more information about the topic.

101: Shortwave radiation, which includes direct radiation and diffuse radiation

108: You might consider quoting Birnbaum et al. 2011 here (she is one of your co-authors anyway).

111/112: is that your own definition of snowfall and light snowfall? Does "light snowfall" correspond to diamond dust? (Be careful with terms that are already defined)

122: "ratio" rather than relative composition

126: dito

136: explain SLAP and GISP

162: relationship between ïĄďD and q

222/224: snow accumulation or erosion (not depletion)

243: what is the approximate height of the lowest model level? How is the model 2m-temperature calculated? This would help to explain the differences between model and observations.

256: see above

258: hourly values

268: which have data gaps larger than 8h

271: daily mean values of what?

292: strong (or high) depletion

294: you compare only to the simulated data, how about the measured meteorological data?

305: see comment 352

308: "and" rather than "versus"

336: the mean wind speed is... (delete "concerning the wind speed")

352: is that coincidence then or do you have an explanation for it? You mention this several times, and it is not clear if it is just by chance or if it hints at the ability of the model to calculate this correctly.

407: condensation: strictly spoken, condensation is the transition from vapor to liquid, whereas the transition from vapor to solid is called "deposition" according to the Glossary of Meteorology of the AMS (American Meteorological Society). Sometimes the term re-sublimation is used, too. You should explain this at the first point where

"condensation" occurs in the paper and then stick to one expression, whichever you prefer.

413: Fig. 5: surface height anomaly

452: wind advection? There is advection of warm/cold air or moisture, but not of wind. The wind is rather the cause of the advection.

468: please give a reference for the height of the polar boundary layer

472: Do you mean negative and positive trends rather than decreasing and increasing (thus changing) trends?

486: Why do you assume liquid water? Is this only a model assumption or do you have physical evidence for it? Maybe rephrase a bit.

489: with the relative humidity RH. . .. and the kinetic fractionation factor k. . .

536: which is of the same order of magnitude as the observations

551: I suggest deleting "striking". If the results confirmed earlier results by St.-L. they were not too surprising. This does not lower their value, of course.

564: This is likely. . .

570ff: the fractionation takes place no matter if the wind causes erosion or not. Please, rephrase.

576: you write "partly". Are there other possible reasons for the small day-to-day variability?

Fig. 6 - Fig. 9: I suggest removing the figure titles (in the boxes) for clarity, since the legends already take quite a bit of space, and the caption describes the contents anyway. Also inserting grid lines seems to be always helpful. The legend of Fig. 6 would be easier to read if the explanations of the different lines were placed simply to the right of the lines.

Ref.:

Birnbaum, G., J. Freitag, R. Brauner, G. König-Langlo, E. Fischer, S. Kipfstuhl, H. Oerter, C. H. Reijmer, E. Schlosser, S. H. Faria, H. Ries, B. Loose, A. Herber, M. G. Duda, J. G. Powers, K. W. Manning, M. R. Van den Broeke, 2011. Strong wind events and their influence on the formation of snow dunes: Observations from Kohnen Station, Dronning Maud Land. J. Glaciol. 56 (199), 891-902

Schlosser, E., K. W. Manning, J. G. Powers, M. G. Duda, G. Birnbaum, K. Fujita, 2010. Characteristics of high-precipitation events in Dronning Maud Land, Antarctica. J. Geophys. Res., 115, D14107, doi:10.1029/2009JD013410.
* * *

---

## Referee Comment (RC2) · B. Markle (Referee) · 14 Apr 2016

In this study the authors present continuous measurements of the isotopic composition of vapor at an East Antarctic station, an important and novel achievement. Further they demonstrate that the isotopic composition of the surface of the snowpack changes in phase with the vapor, even in the absence of precipitation, over a short interval of study. This important finding had been demonstrated only recently in Greenland and not yet shown in Antarctica to my knowledge. The authors create a usefully simple and elegant box model to understand the influence of the vapor on the snowpack and convincingly explore sensitivities to important parameters.

Many of the authors' findings and conclusions are both novel and important. Their use

of a simple box model to understand the magnitudes of changes in the snow pack due to the vapor is compelling. The main conclusion, that post-depositional processes can significantly alter Antarctic snowpack, has great importance for a number of fields, perhaps most significantly the interpretation of ice core isotopic records for paleoclimate. The study, along with previous work in Greenland, represents a critical first step in what will surely be a productive line of research.

The study also features substantial analysis of isotope-enabled GCM results and comparisons to the observed record. While this analysis is well done and useful, it currently feels out of place in the study, or is at least under-utilized with respect to their discussion, interpretations, conclusions, and implications. This is in contrast to their use of a simple box model, which very clearly contributes to their understanding of physical processes and conclusions.

The writing is generally clear, though it is not without errors and frequently awkward or unusual phrasing, which detracts slightly from the work and occasionally obscures the meaning of a sentence. I noted some, though not all, of these instances below. The figures are generally excellent. From a methodological and practical point of view, the paper will be a useful and influential addition to the literature. In particular, the descriptions of their methodologies, error propagation in their measurements, and the design of their box model are excellent and will be a boon to many future researchers. I'd recommend that this paper be accepted after revisions.

General comments/concerns:

1) The authors make extensive comparisons of their vapor measurements to results from GCMs. These comparisons are well done and useful, though it is not clear how they fit into the overall point of the paper. There is relatively little discussion of the comparisons or their implications. While there is much description of the modeling results, there is very little interpretation. In fact, the simulations are not mentioned in the conclusions at all! Nor in the abstract, nor in the title. Yet the topic represents ~5

pages of the main text. I'm left wondering what the point of this analysis was.

This is a shame, because there is substantial and useful work presented here.

From another point of view, if the reader is going to read a significant amount of text about the GCM simulations and their comparison to observations, they ought to come away with having learned something about their implications.

For example, much discussion is given to the relative performance of the two models against observations. Yet little discussion of the possible source of these differences is given. Is it differences in the isotope schemes in the models? Is it the different re-analysis data used to force (the lower boundary) and nudge the models? Suggestions toward answers to these questions are presented in the text, yet no interpretation is given. While solving these questions is beyond the scope of this study, some discussion is certainly warranted.

I was surprised that no analysis of the isotopic composition of precipitation in the model was made or compared to the mean observed values of the snow surface. How do monthly or daily mean isotopic values of precipitation or weighted accumulation in each model compare against observed mean values of the snow surface? How important are the post-depositional processes that are not represented by the models? That is, how different are the simulated precipitation weighted values to the values during precipitation at the site and to the value of the snow pack that interacts with the vapor over the same period. This comparison would be an excellent illustration of the importance of these findings.

At the very least I think some conclusions about model differences and performance, ability to simulate isotopic changes in vapor, and the importance of not simulating the post depositional processes is warranted. Otherwise it is not at all obvious what the point of including that analysis is.

2) In a related point, the authors quite rightly frame the importance of this work in terms

of the interpretation of deep ice core records. However, aside from the statement that it is important (which it undoubtedly is), little discussion of how or why it is important is made. If one assumed that the snowpack over the observational period represented the weighting of just the precipitation events vs. a snowpack continuously interacting with the vapor, how different would the mean values be? What about in the models? Over what timescales is this likely to be important? Over what depth in the snow might these post-depositional processes be relevant? At what sites in Antarctica might this process be more or less important? Given the episodic nature of snowfall at the site and typical amounts of accumulation in those events, and the depth over which these post-depositional processes operate, what fraction of an annual layer of accumulation at Kohnen station can be thought of as having precipitation-weighted isotopic values vs. vapor-altered isotopic values?

I think discussion of some of the above types of questions, all of which would require only simple calculations from the data the authors have already presented, would greatly enhance the utility and impact of this study, and specifically toward the stated goal of better understanding ice core records.

Further, I think some discussion about the potential limits to the impact of these post depositional effects is also warranted. The snow surface study, through which this process is revealed, represents less than a day and half of time. And this was not a particularly normal day and a half either, showing rather high values of q, and subdued diurnal cycles in several important metrologic parameters, as the authors note. I think some discussion of whether these unusual conditions might contribute (or not) to the post-depositional processes seems useful.

All of the above recommendations ought only to serve to highlight the importance of further studies of this type.

Specific comments and technical corrections (line by line):

line 6: I assume the use of the "synoptic variability" is here meant to refer to the

timescales associated with synoptic events (rather than a spatial scale) given the comparison to the diurnal cycle. Since "synoptic" technically refers to a horizontal length scale in meteorology ($\sim$1000 km), the current wording may slightly confuse the reader in thinking that a comparison is being made to spatial variability of isotopes in vapor. Perhaps simply changing the wording to the following would avoid this small issue: "During our monitoring period, the variability of the water vapor isotopic composition at timescales associated with synoptic events is found to be low compared to the diurnal cycle..."

Line 9: "snow surface" = what depth?

Wording is occasionally awkward throughout the text. Eg. Line 36. "...the mean precipitation isotopic composition..." is slightly confusing and the meaning somewhat ambiguous (what does "mean" apply to? The "mean composition" or the "mean precipitation"?). I assume this means the "mean isotopic composition of precipitation", but if not, the meaning is unclear. There are other similar instances though I've not highlighted them all. Generally these instances do not interfere with the otherwise very clear writing, but they are somewhat distracting.

Line 68: It is unclear what "moisture level" specifically refers to. Specific humidity? Accumulation?

Line 157: What is the "Anderson correction" a correction for?

Line 224: I believe the use of "depletion" here should actually be "ablation" or something equivalent. Unless the authors are actually talking about depletion of isotopes, in which case the meaning is unclear. In either case, please correct or explain in more detail.

Line 229: The authors refer to the "large variability in surface isotopic composition". Is this known previously (if so please cite a relevant reference) or assumed or just potentially present? Please clarify.

Line 230-231: Regarding the qualitative descriptions of the snow surfaces ("hard", "soft", etc): could you briefly state what this is based on? Were these based on real density differences, qualitative assessment, etc? This could be useful information for follow-on studies.

Section 3.6: Is the local weather station at Kohnen used in either of the two reanalysis products?

Lines 245-250: Can you explain why the LMDZ5Aiso is nudged with ECMWF wind fields and forced with NCEP SSTs at the lower boundary? Is there not potential for self-inconsistencies between the winds and temperature gradients?

Line 251: What does "equilibrated" mean precisely in the case of an atmosphere-only, reanalysis-nudged, ∼35 year simulation? This is not obvious. Do the authors just mean "integrated"?

Lines 268-270: Please make clear that you are discussing the observations initially, rather than the simulations. It is not stated nor immediately obvious from the previous paragraph.

Line 276: I don't think "satisfying" is the word you mean. Perhaps "satisfactory"? A quantitative statement about the performance would be better still.

Section 4.1: What is the height/pressure of the first vertical level in the model(s) and what is the near-surface resolution in height/pressure? This is not stated in the methods. Presumably the vapor isotopic values being compared here are from the first vertical level. Thus it is important to know what the level represents physically for comparison to the near surface observations. What is the vertical change in vapor isotopic values across the few bottom-most levels in the model? The presence of strong vertical gradients near the surface in the model may be important to understanding the comparison between model results and data. Please provide this information and perhaps some brief discussion on its relevance (or not) to mismatch between the simulations

and observations.

Line 294: Any sense of what is the source of the strongly depleted events in ECHAM is, if not associated with any particular meteorological variable? Are there potentially numerical issues at very low depletion levels in the model?

Line 296: It is not obvious that "during the night" means much in this context. It is 24hr daylight, no? Is this the diurnal temperature minimum? Just stating the hours seems sufficient.

Line 297: I'm not sure "interfere" is the appropriate word in this context. Perhaps "complicates".

Line 299: "with" should be "to".

Section 4.2: Throughout this section, it is often not immediately clear whether a particular sentence is referring to observations or simulations, e.g. line 379.

Line 366: Please make it more clear which is lagging behind which. Is it the 3m lagging behind the 0.2m?

Line 370-375: It would be appropriate here to remind the reader what the equivalent height the modeled isotope values of vapor are for.

Line 390: ECHAM tends to overestimate slopes compared to what? To observations?

Line 395-405: Several typos.

Line 410: "which" should be "what".

455-460: The conclusion that 40% of vapor mixing ratio variability is sufficient to understand isotope ratio variability is very interesting!

Line 468: Where does the expectation that the polar boundary layer height is ∼50-100m come from? A relevant reference would be useful. And what does this refer to? Is this an e-folding height of moisture content? Is it the height of the well-mixed layer?

Lines 469-475: The order of the cases you describe, i)-290 and ii)-310, are reversed between the figure and the text. This is initially confusing.

Line 479-481: The authors conclude that condensation is "the likely cause" of the observed changes in isotopic composition of the snow surface. The statement "the likely cause" implies that there has been an assessment of the likelihood of several (at least more than one) possible mechanisms to explain these variations and that this particular mechanism is preferred. This may be the case, but the authors have not shown this. Instead they have shown that condensation, which they expect to be happening due to changes in the saturated mixing ratio, can readily explain the observed changes in the surface, within uncertainties in their model. This is a fantastic finding! But there has been no analysis of other possible mechanisms. While surely subtle, and perhaps pedantic, the distinction is important. A slight change in wording is warranted.

Line 530-531: The wording here is awkward and the meaning obscured.

Line 537: Watch subject agreement throughout the text, e.g. in this line "reservoir heights".

Line 540: This is an important finding!

Conclusions: There are some well made and important conclusions here, well done. The phrasing throughout this section (more so than others) is often unfortunately awkward or unusual, which slightly detracts from the otherwise high impact.

Figures: The figures are generally excellent. Very clear and informative. In figures 7 and 9, black bars are used to show the range of the observations. Would it not be more useful to actually show the trends in the observations? Isn't the temporal evolution important and useful for comparison to the model results? Perhaps they have been removed for clarity, but I think their inclusion in at least one panel would be useful.

---

## Author Comment (AC1) · 28 Jun 2016

**Response to anonymous Referee #1**

We would like to thank Referee 1 for helpful review comments. The scientific terms used in the revised manuscript have been better defined to improve clarity. We now provide answers to general comments and then report on detailed comments by quoting the revised text.

■ *It is mostly well written, the English is generally ok, but not free of small errors (tenses etc.) and slightly awkward formulations, on which I won't always comment in detail since I believe that this is not the job of a reviewer (while professional companies charge a lot of money for it). This should be checked.*

None of our coauthors is a native-born English speaker, and we apologize for our awkward formulations. Following your comment, we did our best to improve the readability of our revised manuscript.

■ *The description of previous work could be a bit more elaborate.*

We did improve the description of earlier work done at Dome C and Vostok, Antarctica:

L81: "The surface of the ice sheet around Kohnen is characterized by the presence of large sastrugi, created by wind redistribution and sublimation of snow, hence producing considerable variability in the snow surface age, origin and density. In particular, very hard dunes sticking up above the mean surface level may be half a year old (Birnbaum et al., 2010). A previous study performed at Vostok station (Antarctica) reported a large variability in the isotopic composition of the snow surface (10 cm depth) over an 1 km transect, with a maximum variation of 30‰ in $\delta D$ over 100 m (Ekaykin et al., 2002)."

L597: "The parameter $H_t$ is therefore the mixing-layer height. Sodar measurements performed at Dome C (Antarctica) showed magnitudes between 10 m and 300 m (Pietroni et al., 2012; Casasanta et al., 2014)."

**Line by Line comments**

**Major comments**

■ *32: what is a meteorological time scale? Better use "synoptic" if that is what you mean. (e.g. diurnal variations are also meteorological)*

L40: "only few studies have examined the drivers of Antarctic precipitation isotopic composition variability at timescales associated with synoptic or diurnal events"

■ *45: to avoid any confusion, please define humidity mixing ratio*

L56: "humidity mixing ratio (defined as the mass of water vapor divided by the mass of dry air)"

■ *91ff: How do you define a precipitation event? It is not correct that precipitation occurs only in summer, and at Kohnen station, only about 20% of snowfall events is directly related to synoptic systems. (additionally, there is diamond dust precipitation, which is not so rare). I personally don't like reviewers, who always want their own papers quoted, but, of course, everybody knows their own papers best and, unfortunately, I do not know any other paper that investigates Kohnen precipitation but Schlosser et al. (2010), which gives some more information about the topic.*

Snowfall and light snowfall events are defined in section 3.1.
L115: "The katabatic regime can be interrupted by the influence of synoptic systems, responsible for 20 % of snowfall events at Kohnen station (Schlosser et al., 2010)."

■ *108: You might consider quoting Birnbaum et al. 2011 here (she is one of your coauthors anyway).*

The description of the snow surface texture has been moved to the end of the introduction.
L84: "In particular, very hard dunes sticking up above the mean surface level may be half a year old (Birnbaum et al., 2010)."

■ *111/112: is that your own definition of snowfall and light snowfall? Does "light snowfall" correspond to diamond dust? (Be careful with terms that are already defined)*

L138: *"A snowfall event leaves a visible accumulation on flat surfaces (for example transport boxes) whereas during a light snowfall event or a diamond dust event no visible accumulation is observed."*

■ *243: what is the approximate height of the lowest model level? How is the model 2m temperature calculated? This would help to explain the differences between model and observations.*

L285: "The lowest model level (about 60 m above the surface) has been selected followed […]"
L309: "Furthermore, at Kohnen Station, the 2 m temperature in ECHAM5-wiso is calculated from the surface energy balance equation, assuming a constant surface albedo of 0.8"

■ *256: see above*

L305: "However, the reader should notice that simulated parameters such as humidity or water vapor isotopes are extracted from the first vertical model level (which represents a height of 60 m above ground) whereas the in-situ observations are measured close to the surface."

■ *271: daily mean values of what?*

L326: *"29 daily mean values of q, T, d-excess and δD have been calculated for the observations and the model outputs."*

■ *294: you compare only to the simulated data, how about the measured meteorological data?*

The highly depleted isotopic values simulated by ECHAM5-wiso arise from a computing issue that is certainly not related to a real meteorological event. The description of this issue has been improved to avoid misunderstanding:

L364: "These depletion events do not correspond to any parallel signal in the simulated meteorological data (cloud cover, wind speed, temperature or humidity mixing ratio) and further analyses will be necessary to understand these artifacts."

■ *352: is that coincidence then or do you have an explanation for it? You mention this several times, and it is not clear if it is just by chance or if it hints at the ability of the model to calculate this correctly. {305: see comment 352}*

L381: *"This might be explained by a stronger dependency of deuterium excess variability to climate conditions during evaporation processes in the vapor source regions, while the δD signal is understood to be more directly controlled by climate conditions near or at Kohnen station."*

■ *407: condensation: strictly spoken, condensation is the transition from vapor to liquid, whereas the transition from vapor to solid is called "deposition" according to the Glossary of Meteorology of the AMS (American Meteorological Society). Sometimes the term re-sublimation is used, too. You should explain this at the first point where "condensation" occurs in the paper and then stick to one expression, whichever you prefer.*

This is a very good point, not properly addressed in the original manuscript.

L91: "The term "condensation" (rather than "deposition") is preferred in this paper to describe the water phase change from vapor to solid in order to avoid a possible confusion with "post-depositional processes"."

■ *452: wind advection? There is advection of warm/cold air or moisture, but not of wind. The wind is rather the cause of the advection.*

L577: *"In reality, there is advection of air masses with different moisture or temperature into and out of the box."*

■ *468: please give a reference for the height of the polar boundary layer*

L597: "The parameter $H_t$ is therefore the mixing-layer height. Sodar measurements performed at Dome C (Antarctica) showed magnitudes between 10 m and 300 m (Pietroni et al., 2012; Casasanta et al., 2014)."

■ *472: Do you mean negative and positive trends rather than decreasing and increasing (thus changing) trends?*

L605: *"In the first case, the mixing between the condensate and the snow surface will tend toward the equilibrium through a positive trend; in the second case, a negative trend is predicted."*

■ *486: Why do you assume liquid water? Is this only a model assumption or do you have physical evidence for it? Maybe rephrase a bit.*

The paragraph describing the different hypotheses of fractionation occurring during sublimation has been rephrased:

L617: *"It is generally assumed that no fractionation occurs during sublimation. Using Greenland data, Steen-Larsen et al. (2011) and Landais et al. (2012) showed that on average the snow surface isotopes and the water vapor isotopes are in equilibrium, and estimated that the value of the equilibrium factor lies between the fractionation coefficient $\alpha_{ice}$ with respect to ice (Merlivat and Nief, 1967; Ellehoj et al., 2013) and the fractionation coefficient $\alpha_{water}$ with respect to water (Majoube, 1971). In this study, we test different hypotheses to obtain a range of prediction of the isotopic variation in the vapor and the snow surface."*

■ *551: I suggest deleting "striking". If the results confirmed earlier results by St.-L. they were not too surprising. This does not lower their value, of course.*

The word "striking" has been removed.

L705: *"As these variations in the surface snow isotopic composition follow the diurnal trend in the air, this result confirms the observations of Steen-Larsen et al. (2013) at NEEM"*

■ *570ff: the fractionation takes place no matter if the wind causes erosion or not. Please, rephrase.*

The fractionation does indeed take place with or without wind erosion, however the wind drift could bring ice crystals coming from another source.

L726: "We do observe an increase in $\delta D_s$ during the sublimation process, which indicates that water isotopes undergo fractionation during sublimation. The only doubt we could emit on this result is related to the wind drift. Effectively, the isotopic variability observed on the diurnal scale in a snow patch could also be attributed to the renewal of the snow surface by the wind, which mixes the surface of the snow with ice crystals coming from other snow patches."

■ *576: you write "partly". Are there other possible reasons for the small day-to-day variability?*

No other reasons were possible, the word "partly" has been removed.

■ *Fig. 6 - Fig. 9: I suggest removing the figure titles (in the boxes) for clarity, since the legends already take quite a bit of space, and the caption describes the contents anyway. Also inserting grid lines seems to be always helpful. The legend of Fig. 6 would be easier to read if the explanations of the different lines were placed simply to the right of the lines.*

The figure titles have been removed. In order to allow readability of the figures which already have a lot of material, and because the key message is not in the exact values but on the magnitude of variations, we have decided not to insert additional grid lines in Fig. 6-9.

**Minor comments, all corrected**

*101: Shortwave radiation, which includes direct radiation and diffuse radiation* → L126

*122: "ratio" rather than relative composition* → L153

*136: explain SLAP and GISP* → L167

*162: relationship between d18O, dD and q* → L196

*222/224: snow accumulation or erosion (not depletion)* → L256

*258: hourly values* → this paragraph has been removed

*268: which have data gaps larger than 8h* → 325

*292: strong (or high) depletion* → L361

*308: "and" rather than "versus"* → L385

*336: the mean wind speed is… (delete "concerning the wind speed")* → L421

*413: Fig. 5: surface height anomaly* → done

*489: with the relative humidity RH... and the kinetic fractionation factor k...* → L630

*536: which is of the same order of magnitude as the observations* → L681

*564: This is likely....* → L719

---

## Author Comment (AC2) · 28 Jun 2016

**Response to referee #2, Bradley Markle**

Dear Bradley Markle,

We would like to thank you for your contribution to this manuscript which, we hope, is now far better. Let's now immediately jump to your general comments and concerns:

■ *1) The authors make extensive comparisons of their vapor measurements to results from GCMs. These comparisons are well done and useful, though it is not clear how they fit into the overall point of the paper. There is relatively little discussion of the comparisons or their implications. While there is much description of the modeling results, there is very little interpretation. In fact, the simulations are not mentioned in the conclusions at all! Nor in the abstract, nor in the title. Yet the topic represents 5 pages of the main text. I'm left wondering what the point of this analysis was. This is a shame, because there is substantial and useful work presented here. From another point of view, if the reader is going to read a significant amount of text about the GCM simulations and their comparison to observations, they ought to come away with having learned something about their implications. For example, much discussion is given to the relative performance of the two models against observations. Yet little discussion of the possible source of these differences is given. Is it differences in the isotope schemes in the models? Is it the different reanalysis data used to force (the lower boundary) and nudge the models? Suggestions toward answers to these questions are presented in the text, yet no interpretation is given. While solving these questions is beyond the scope of this study, some discussion is certainly warranted. I was surprised that no analysis of the isotopic composition of precipitation in the model was made or compared to the mean observed values of the snow surface. How do monthly or daily mean isotopic values of precipitation or weighted accumulation in each model compare against observed mean values of the snow surface? How important are the post-depositional processes that are not represented by the models? That is, how different are the simulated precipitation weighted values to the values during precipitation at the site and to the value of the snow pack that interacts with the vapor over the same period. This comparison would be an excellent illustration of the importance of these findings. At the very least I think some conclusions about model differences and performance, ability to simulate isotopic changes in vapor, and the importance of not simulating the post depositional processes is warranted. Otherwise it is not at all obvious what the point of including that analysis is.*

Thank you for this comment, we have taken it into account. We substantially reworked the manuscript to include a discussion of the AGCM outputs and interpretation of the model behavior.

Let's detail here the changes made on the new manuscript with respect to the AGCMs:

- The abstract has been reworked, and the following paragraph has been added:

L6: "Observations have been compared with the outputs of two atmospheric general circulation models (AGCMs) equipped with water vapor isotopes: ECHAM5-wiso and LMDZ5Aiso. During our monitoring period, the signals in the 2 m air temperature T, humidity mixing ratio q and both water vapor isotopes dD and d18O are dominated by the presence of diurnal cycles. Both AGCMs simulate similar diurnal cycles with an amplitude 30 % to 70 % lower compared to the observations, possibly due to an incorrect simulation of the surface energy balance and the boundary layer dynamic."

- The section 3.6 describing the AGCMs is now much more specific and contains information on the first grid height and reanalyzes.
- Two paragraphs on the surface energy balance simulated by the AGCMs have been added, with a focus on the surface temperature:

L346: "We notice that the mean radiative input (in longwave and shortwave) measured at the surface by the AWS 9 is 583 W.m$^{-2}$, compared to only 552 W.m$^{-2}$ for LMDZiso. An incorrect simulation of the cloud cover (and subsequently the precipitation) is likely related to this offset in LMDZiso. The surface energy balance determines the mean surface temperature (-27 °C for LMDZiso compared to -24 °C for ECHAM5-wiso), which itself impacts the sublimation rate and the 2 m air temperature via sensible and latent heat exchanges with the lower atmosphere. The radiative offset present in LMDZiso could explain the low simulated values of temperature and humidity mixing ratio."

L438: "Both models underestimate the amplitude of the diurnal cycle in the air temperature at 2 m by more than 50 %. The surface temperature simulated by both AGCMs has a peak-to-peak amplitude of 7 °C, compared to 14 °C for the measurements of Van As et al. (2005). Variations of the surface temperature at Kohnen are supposed to be driven on the first order by the radiative budget. We have therefore compared the radiative budget of the AWS 9, ECHAM5-wiso and LMDZiso. Both models show good agreement with the observations for the net shortwave budget at the surface. However, the longwave radiative components are more difficult to simulate. Downward longwave emissions are related to the cloud cover (greenhouse effect) and snowfalls, whereas upward longwave emissions are related to the surface temperature and emissivity of the surface. Both models show difficulties simulating a proper cloud cover and snowfall events, and the variation in their surface temperature is 50 % lower than observed. That explains the disagreement between the observations and the AGCMs with respect to the longwave radiative budget, leading to a wrong simulation of the surface temperature."

- We manage to prove that the AGCMs are not able to simulate katabatic winds:

L421: "Both models fail to simulate the pattern of katabatic winds. The mean wind direction is 32 ± 27 ° for ECHAM5-wiso and 21 ± 24 ° for LMDZiso, and their mean wind velocity at 10 m is only 3.1 m.s-1 for ECHAM5-wiso and 1.9 m.s-1 for LMDZiso. They also show a low diurnal variability, whereas Van As et al. (2005) observed at the same height variations higher than 2

m.s$^{-1}$ over 24 h. The underestimation might be due to the horizontal resolution, which is too coarse to represent properly the katabatic winds, especially in LMDZiso."

- We explain why we do not manage to analyze the boundary layer dynamic simulated by AGCMs:

L461: "The height and stability of the boundary layer is particularly difficult to simulate over ice, and have a certain impact on the presence or absence of diurnal cycles (Holtslag et al., 2013). A proper understanding of the simulation of the boundary layer by the AGCMs would require relevant output parameters such as the boundary layer depth or stability classes, which have not been implemented yet. Further analyses will therefore be necessary to understand the different behavior of LMDZiso and ECHAM5-wiso."

- We do not compare the isotopic composition of the precipitation simulated by the AGCMs because we did not collect precipitation samples at Kohnen. However we can compare the isotopic composition of the snow surface simulated by the AGCMs with our snow surface samples:

L517: "The mean deuterium value of the snow patches varies from -296 ‰ to -316 ‰, showing that the texture of the snow patch and its isotopic composition could be related (Table 6). This observation confirms the spatial variability previously observed at Vostok in the isotopic composition of the snow surface (10 cm depth), with variations up to 30 ‰ in dD over 100 m horizontally (Ekaykin et al., 2002). Both AGCMs manage to simulate a similar isotopic composition for the snow surface, with on average a deuterium value of -330 ‰ for ECHAM5-wiso and -299 ‰ for LMDZiso. As expected, the isotopic composition of the snow surface simulated by the AGCMs depends on the snowfall events only, with a variation in dD of 6 ‰ for ECHAM5-wiso over the study period (no variation is simulated by LMDZiso)."

- A new paragraph dedicated to the AGMCs has been added to the conclusion:

L694: "Outputs from the two AGCMs (ECHAM5-wiso and LMDZiso) show in general good agreements with the observations. However, the surface temperature variations simulated by the models have an amplitude 50 % lower than observed by Van As et al. (2005), likely due to the difficulty to simulate the longwave radiative budget (related to the cloud cover and snowfall events). Moreover, the strong katabatic winds observed at Kohnen are not properly simulated by the AGCMs. The simulation of processes in the polar boundary layer and associated inversion is also known to be a challenge for AGCMs (Holtslag et al., 2013). This could explain why the amplitude of the diurnal cycles is lower in the models compared to the observations."

We hope that these changes will be sufficient to justify the presence of AGCM simulations in our article. Further work will be required to understand the boundary layer dynamic simulated by the AGCMs, and subtle differences between LMDZiso and ECHAM5-wiso. As you mention it, this is "*beyond the scope of this study*", because the main focus of this paper is to describe and understand the isotopic exchange between the snow surface and the lower atmosphere.

■ *2) In a related point, the authors quite rightly frame the importance of this work in terms of the interpretation of deep ice core records. However, aside from the statement that it is important (which it undoubtedly is), little discussion of how or why it is important is made. If one assumed that the snowpack over the observational period represented the weighting of just the precipitation events vs. a snowpack continuously interacting with the vapor, how different would the mean values be? What about in the models? Over what timescales is this likely to be important? Over what depth in the snow might these post-depositional processes be relevant? At what sites in Antarctica might this process be more or less important? Given the episodic nature of snowfall at the site and typical amounts of accumulation in those events, and the depth over which these post-depositional processes operate, what fraction of an annual layer of accumulation at Kohnen station can be thought of as having precipitation-weighted isotopic values vs. vapor-altered isotopic values? I think discussion of some of the above types of questions, all of which would require only simple calculations from the data the authors have already presented, would greatly enhance the utility and impact of this study, and specifically toward the stated goal of better understanding ice core records. Further, I think some discussion about the potential limits to the impact of these post depositional effects is also warranted. The snow surface study, through which this process is revealed, represents less than a day and half of time. And this was not a particularly normal day and a half either, showing rather high values of q, and subdued diurnal cycles in several important meteorological parameters, as the authors note. I think some discussion of whether these unusual conditions might contribute (or not) to the post-depositional processes seems useful. All of the above recommendations ought only to serve to highlight the importance of further studies of this type.*

We do agree on the importance of quantifying the impact of post depositional processes on ice core data. However, in our opinion, this matter is beyond our present skills and simplistic calculations would not make sense for the two following reasons:

I) The box model we have developed is a closed system, with an exchange of water molecules between two reservoirs occurring during condensation and sublimation. Despite the simplicity of this model, the input parameters are difficult to constrain (e.g. the size of the reservoirs, the fractionation coefficients or the initial isotopic value of the snow pack). The wide range of simulated values obtained with this model does not allow us to extend the results to Antarctica. Even if the model outputs would be specific and the model well constrained, the reality is much more complex. The advection of air masses should be implemented (strong kabatic diurnal cycles), as well as possible exchanges with the free troposphere (the mixing-layer height is

known to vary from 10 m to 300 m, e.g., Pietroni et al., 2012). The snow is also known as a porous material, with an important spatial variability in its density, texture and isotopic composition (Ekaykin et al., 2002).

II) Diurnal exchanges do not explain the annual isotopic variability. Synoptic events must be considered if we want to understand what is happening in the snow surface in terms of isotopic variation. In this paper, we have only studied the diurnal scale because of the quasi-absence of a synoptic signal during the study period. A simple calculation based on our observations would be the following: if the diurnal cycle observed in the isotopic composition of the snow surface is symmetrical, then the daily isotopic change is expected to be null in the snow surface. Taking into account the uncertainties associated with the snow sampling protocol, we cannot assess whether or not the net isotopic budget of the snow surface over 24 h is different from zero.

The purpose of this article is to prove that continuous water vapor isotopic measurements are technically possible in Antarctica and to show also the existence of an isotopic exchange between the near surface snow and the lower atmosphere on the diurnal scale. The simple box model developed in our article is the first step to understand how post depositional processes operate. We prefer to wait for further field experiments, reproduction of our protocol and improvement of the model before studying the impact of post depositional processes on ice core data.

However, we can answer to two questions:

*Over what depth in the snow might these post-depositional processes be relevant?*

L724: "According our box model, no diurnal cycle in the isotopic composition of the snow surface is expected from a depth of 1 cm or above."

*Over what timescales is this likely to be important?*

Changes in the isotopic composition of the snow surface are observed over 12 h. Our article proves that the post-depositional process occurs on an hourly time scale.

**Line by Line comments**

**Major comments**

■ *line 6: I assume the use of the "synoptic variability" is here meant to refer to the timescales associated with synoptic events (rather than a spatial scale) given the comparison to the diurnal cycle. Since "synoptic" technically refers to a horizontal length scale in meteorology (1000 km), the current wording may slightly confuse the reader in thinking that a comparison is*

*being made to spatial variability of isotopes in vapor. Perhaps simply changing the wording to the following would avoid this small issue: "During our monitoring period, the variability of the water vapor isotopic composition at timescales associated with synoptic events is found to be low compared to the diurnal cycle..."*

The abstract has been rephrased, the word "synoptic" does not appear anymore.

■ *Line 9: "snow surface" = what depth?*

L12: "In parallel, snow surface samples were collected each hour during 35 h, with a sampling depth of 2-5 mm."

■ *Line 36. "...the mean precipitation isotopic composition..." is slightly confusing and the meaning somewhat ambiguous (what does "mean" apply to? The "mean composition" or the "mean precipitation"?). I assume this means the "mean isotopic composition of precipitation"*

L46: "Classically, the mean isotopic composition of precipitation simulated by atmospheric models is directly compared to ice core data,"

■ *Line 68: It is unclear what "moisture level" specifically refers to. Specific humidity? Accumulation?*

L78: "These measurements were performed at the German Kohnen station, a deep ice coring site with intermediate temperature and a humidity mixing ratio high enough in the summer for making accurate measurements of the water vapor isotopic composition."

■ *Line 157: What is the "Anderson correction" a correction for?*

L189: "The humidity mixing ratio is calibrated against the relative humidity measured by the AWS9. This relative humidity has been previously calibrated following the protocol of Anderson (1994), setting its maximum values equal to 100 % of humidity."

■ *Line 224: I believe the use of "depletion" here should actually be "ablation" or something equivalent. Unless the authors are actually talking about depletion of isotopes, in which case the meaning is unclear. In either case, please correct or explain in more detail.*

We decided to use erosion instead of ablation.
L256: "In order to detect any snow accumulation or erosion (due to snowfall events or wind drift), 100 thin wood sticks were distributed every meter along a 100 m transect in a clean area and daily measured with a folding ruler. No accumulation or erosion was detected within a precision of 1 mm."

*.■ Line 229: The authors refer to the "large variability in surface isotopic composition". Is this known previously (if so please cite a relevant reference) or assumed or just potentially present? Please clarify.*

We have reworked the end of the introduction and detailed previous studies of the variability in the isotopic composition of the snow surface.

L81: "The surface of the ice sheet around Kohnen is characterized by the presence of large sastrugi, created by wind redistribution and sublimation of snow, hence producing considerable variability in the snow surface age, origin and density. In particular, very hard dunes sticking up above the mean surface level may be half a year old (Birnbaum et al., 2010). A previous study performed at Vostok station (Antarctica) reported a large variability in the isotopic composition of the snow surface (10 cm depth) over an 1 km transect, with a maximum variation of 30‰ in dD over 100 m horizontally (Ekaykin et al., 2002)."

*■ Line 230-231: Regarding the qualitative descriptions of the snow surfaces ("hard", "soft", etc): could you briefly state what this is based on? Were these based on real density differences, qualitative assessment, etc? This could be useful information for follow-on studies.*

This is a very good remark, we have improved the description of the snow sampling protocol.

L263: "Keeping in mind the possible variability in the isotopic composition of the snow surface, three different areas with consistent surface snow texture were selected, based on visual observation (the border of the snow patch was visible) and subjective assessment of the hardness. The snow sampling protocol is based on the assumption that the isotopic composition of a snow patch at a given time is homogeneous. Patch 1 was made of hard ice, patch 2 of compact snow and patch 3 was composed of soft snow. Five adjacent samples for each patch were sampled every hour (15 samples per hour) during a 35-hour period, from 2014/01/08 to 2014/01/10 (as it is shown in Fig. 2 indicated by the SSDC label). The sample depth is estimated between 2 and 5 millimeters, the tool used was a cake spatula."

*■ Section 3.6: Is the local weather station at Kohnen used in either of the two reanalysis products?*

We have not found this information yet, unfortunately.

*■ Lines 245-250: Can you explain why the LMDZ5Aiso is nudged with ECMWF wind fields and forced with NCEP SSTs at the lower boundary? Is there not potential for self-inconsistencies between the winds and temperature gradients?*

L298: "There could theoretically be some inconsistencies between the winds and the SSTs from different reanalyses datasets, but the impact should be very small due to the overall consistency between the two reanalyses datasets and due to the strong nudging of the winds, preventing any drift."

■ *Line 251: What does "equilibrated" mean precisely in the case of an atmosphere-only, reanalysis-nudged, 35 year simulation? This is not obvious. Do the authors just mean "integrated"?*

The word "equilibrated" has been removed.
L286: *"The simulation has been started in 1979 and any potential model spin-up bias, e.g. caused by the initialization of the atmosphere in terms of humidity and its isotopic composition, can be safely neglected for our study period."*
L302: *"Because such high-resolution simulation is costly, the simulation has been started in January 2013 but inspection of simulated time series show that the spinup in sufficient."*

■ *Lines 268-270: Please make clear that you are discussing the observations initially, rather than the simulations. It is not stated nor immediately obvious from the previous paragraph.*

L324: "In order to estimate the magnitude of the day-to-day variability, days with data gaps larger than 8 h have been removed from the data-set (on 12/16, 12/28, 12/29, 01/13, 01/14, 01/17, 01/18 and 01/21) and then 29 daily mean values of q, T, d-excess and dD have been calculated for the observations and the model outputs."

■ *Line 276: I don't think "satisfying" is the word you mean. Perhaps "satisfactory"? A quantitative statement about the performance would be better still.*

L338: "(therefore below the limit of confidence of our instrument, 500 ppmv)"

■ *Section 4.1: What is the height/pressure of the first vertical level in the model(s) and what is the near-surface resolution in height/pressure? This is not stated in the methods. Presumably the vapor isotopic values being compared here are from the first vertical level. Thus it is important to know what the level represents physically for comparison to the near surface observations. What is the vertical change in vapor isotopic values across the few bottom-most levels in the model? The presence of strong vertical gradients near the surface in the model may be important to understanding the comparison between model results and data. Please provide this information and perhaps some brief discussion on its relevance (or not) to mismatch between the simulations and observations.*

In section 3.6, we have indicated the height of the first level grid:

L285: "The lowest model level (about 60 m above the surface) has been selected followed […]"

And made a clear distinction between the measurement height and the height of the simulated outputs:

L304: "Three selected outputs from both models are calculated at a specific height, 10m for the wind speed and wind direction, and at 2 m for temperature. However, the reader should notice that simulated parameters like humidity or water vapor isotopes come from the first vertical model level (which represents a height of 60 m above ground) whereas the in-situ observations are close to the surface. Furthermore, at Kohnen Station, the 2 m temperature in ECHAM5-wiso is calculated from the surface energy balance equation, assuming a constant surface albedo of 0.8. This might also lead to further differences between simulation results and observations."

With a reminder in section 4.2:

L459: "The model-data comparison is hampered by the fact that the simulated humidity mixing-ratio is only available from the first grid level of the AGCMs, and is therefore an average value over the first 60 m. The height and stability of the boundary layer is particularly difficult to simulate over ice, and have a certain impact on the presence or absence of diurnal cycles (Holtslag et al., 2013). A proper understanding of the simulation of the boundary layer by the AGCMs would require relevant output parameters such as the boundary layer depth or stability classes, which have not been implemented yet. Further analyses will therefore be necessary to understand the different behavior of LMDZiso and ECHAM5-wiso."

We believe that a more in depth comparison between isotopic observations and simulations would require isotopic measurements above 10 meters, as it has been performed by Steen-Larsen et al. (2013), and a specific study of the boundary layer dynamic. However, the first variable to study on a vertical scale should be the humidity mixing ratio before water vapor isotopes. This could be the purpose of another article using weather balloons measurements.

■ *Line 294: Any sense of what is the source of the strongly depleted events in ECHAM is, if not associated with any particular meteorological variable? Are there potentially numerical issues at very low depletion levels in the model?*

We suspect numerical issues, but we did not manage to understand where they could come from.

L364: "These depletion events do not correspond to any parallel signal in the simulated meteorological data (cloud cover, wind speed, temperature or humidity mixing ratio) and further analyses will be necessary to understand these artifacts."

■ *Line 296: It is not obvious that "during the night" means much in this context. It is 24hr daylight, no? Is this the diurnal temperature minimum? Just stating the hours seems sufficient.*

This paragraph has been removed.

■ *Section 4.2: Throughout this section, it is often not immediately clear whether a particular sentence is referring to observations or simulations, e.g. line 379.*

We always start by describing the observations, and then we explicitly use the terms "AGCMs" , "ECHAM5-wiso", "LMDZiso", "models" or "simulated values" to make a clear distinction with the simulations.

■ *Line 366: Please make it more clear which is lagging behind which. Is it the 3m lagging behind the 0.2m?*

L477: "the top inlet presents a clear lag of 1 hour behind the bottom inlet with respect to the measured humidity mixing ratio."

■ *Line 370-375: It would be appropriate here to remind the reader what the equivalent height the modeled isotope values of vapor are for.*

L459: "The model-data comparison is hampered by the fact that the simulated humidity mixing-ratio is only available from the first grid level of the AGCMs, and is therefore an average value over the first 60 m".

■ *Line 390: ECHAM tends to overestimate slopes compared to what? To observations?*

L502: "These relationships are better simulated for the diurnal cycle, but ECHAM5-wiso tends to overestimate the associated slopes compared to the observations."

■ *Line 468: Where does the expectation that the polar boundary layer height is 50-100m come from? A relevant reference would be useful. And what does this refer to? Is this an e-folding height of moisture content? Is it the height of the well-mixed layer?*

Thank you for this remark, we now make a distinction between boundary layer and mixing layer.

L597: "The parameter $H_t$ is therefore the mixing-layer height. Sodar measurements performed at Dome C (Antarctica) showed magnitudes between 10 m and 300 m (Pietroni et al., 2012; Casasanta et al., 2014)."

■ *Line 479-481: The authors conclude that condensation is "the likely cause" of the observed changes in isotopic composition of the snow surface. The statement "the likely cause" implies that there has been an assessment of the likelihood of several (at least more than one) possible mechanisms to explain these variations and that this particular mechanism is preferred. This may be the case, but the authors have not shown this. Instead they have shown that condensation, which they expect to be happening due to changes in the saturated mixing ratio, can readily explain the observed changes in the surface, within uncertainties in their model. This is a fantastic finding! But there has been no analysis of other possible mechanisms. While surely subtle, and perhaps pedantic, the distinction is important. A slight change in wording is warranted.*

The word "likely" has been removed.

L613: "However, we are able to conclude that the condensation of water vapor has an effect on the isotopic composition of the top 2 mm of the snow surface."

■ *Line 530-531: The wording here is awkward and the meaning obscured.*

This paragraph has been rephrased.

L672: "If $F_{vt} \neq 0$ (subsaturation), the isotopic composition of the snow is affected by the isotopic composition of the vapor, and in that case the variation of the isotopic composition of a snow patch during the warming phase will depend on its initial isotopic composition (Fig. 7)."

■ *Figures: The figures are generally excellent. Very clear and informative. In figures 7 and 9, black bars are used to show the range of the observations. Would it not be more useful to actually show the trends in the observations? Isn't the temporal evolution important and useful for comparison to the model results? Perhaps they have been removed for clarity, but I think their inclusion in at least one panel would be useful.*

We have added the observed trend in Fig. 7 and Fig. 9.

**Minor comments**

*Line 297: I'm not sure "interfere" is the appropriate word in this context. Perhaps "complicates".* → Paragraph removed

*Line 299: "with" should be "to".* → Paragraph removed

*Line 395-405: Several typos.* → Paragraph rephrased, L509-529

*Line 410: "which" should be "what".* → L534

*Lines 469-475: The order of the cases you describe, i)-290 and ii)-310, are reversed between the figure and the text. This is initially confusing.* → L602

*Line 537: Watch subject agreement throughout the text, e.g. in this line "reservoir heights".* → The subject is plural in this context. L681: "We notice that the uncertainties related to the heights of the reservoirs have…"